# Unsupervised combinatorial optimization under complex conditions: Principled objectives and incremental greedy derandomization

## Abstract

Combinatorial optimization (CO) has significant theoretical and practical implications. CO problems are naturally discrete, making typical machine-learning techniques based on differentiable optimization inapplicable. Karalias & Loukas (2020) adapted the probabilistic method, an important tool in combinatorics, to incorporate CO problems into differentiable optimization. Their work ignited the research on unsupervised learning for CO, composed of two main components: probabilistic objectives and derandomization. Several desirable properties of probabilistic objectives have been proposed, but without principled schemes to satisfy them. Also, the derandomization process is still underexplored. Motivated by the limitations, we propose our method UCom2, consisting of two schemes: (1) a *principled* probabilistic objective construction scheme that provably satisfies the good properties, and (2) a *fast* and *effective* derandomization scheme with a quality guarantee. We apply UCom2 to various *complex conditions* (e.g., cardinality constraints, non-binary decisions) and problems involving them, highlighting that UCom2 is *general* and *practical*. We further show the empirical superiority of UCom2 w.r.t. both optimization quality and speed.

## 1 Introduction

Combinatorial optimization (CO) is a branch of optimization. Typically, among a discrete set of choices, a CO problem aims to find the best one that minimizes (or maximizes) some objectives. CO is theoretically important as a mathematical subfield. Many CO problems are NP-hard (Papadimitriou & Steiglitz, 1998; Karp, 2010), and investigating CO problems helps us study computational complexity and many other related fields like cryptography (Applebaum et al., 2010). Also, many CO problems are derived from real-world applications (Paschos, 2014).

Naturally, CO problems are *discrete*. Specifically, the space of feasible solutions is discrete, and the objective cannot be directly evaluated outside the discrete space. While recent years have witnessed rapid developments in machine learning (ML), most ML methods are based on differentiable optimization (e.g., gradient descent). Thus, applying such ML techniques to CO problems is non-trivial. In their pioneering work, Karalias & Loukas (2020) adapted the probabilistic method (Erdős & Spencer, 1974; Alon & Spencer, 2016), an important tool in combinatorics, to incorporate discrete CO problems into differentiable optimization. Specifically, they proposed to evaluate CO objectives on a *distribution* of discrete choices (i.e., in a *probabilistic* manner). This allows continuous parameterization such that applying ML techniques with differentiable optimization becomes feasible. In addition to differentiable optimization, their pipeline consists of two main steps: (1) construction of *probabilistic objectives* (i.e., a continuous version of the original optimization objectives); (2) *derandomization* to obtain the final discrete solutions. This ignited the line of research on unsupervised (i.e., not supervised by solutions) learning for combinatorial optimization (UL4CO).

However, the prior works on UL4CO share multiple limitations. Although Karalias & Loukas (2020) and Wang et al. (2022) proposed some desirable properties of the probabilistic objectives, they did not propose any *practical* schemes to construct the objectives with such properties. At the same time, the derandomization process is underexplored, without many practical techniques or theoretical discussions. Particularly, we do not have guidelines for UL4CO under *complex conditions* (i.e., complex optimization objectives and/or constraints) that are mathematically hard to handle.

Motivated by the limitations, we propose UCOM2 (**U**nsupervised **Com**binatorial Optimization **U**nder **Com**plex Conditions), composed of a probabilistic objective construction scheme and a derandomization scheme. UCOM2 has the following theoretical and empirical strengths:

- **Probabilistic objective construction scheme:** UCOM2 is equipped with the first *principled* scheme for constructing probabilistic objectives, satisfying the good properties suggested by Karalias & Loukas (2020) and Wang et al. (2022). Specifically, we show that any probabilistic objective that can be rephrased as an expectation would suffice in satisfying the properties (Sec. 3.1).
- **Derandomization scheme:** UCOM2 is equipped with a *fast* and *effective* derandomization scheme. We show that, when a probabilistic objective is constructed following the first scheme, the derandomized results have a stronger quality guarantee than the existing ones (Sec. 3.2).
- **Applicable to complex conditions:** UCOM2 is *general* and *practical*. We apply UCOM2 to complex conditions (e.g., cardinality constraints and non-binary decisions; Sec. 4). Further, for CO problems involving such complex conditions, we complete objective construction and derandomization by combining our derivations on their corresponding conditions (Sec. 5).
- **Experiments**: UCOM2 has empirical superiority w.r.t. optimization quality and speed (Sec. 6).

## 2 PRELIMINARIES AND BACKGROUND

### 2.1 MATHEMATICAL PRELIMINARIES AND NOTATIONS

**Graphs.** A *(edge-weighted) graph* $G = (V, E, W)$ is defined by a *node set* $V$, an *edge set* $E$, and *edge weights* $W : E \to \mathbb{R}$. We let $n = |V|$ denote the number of nodes (WLOG, $V = [n] := \{1, 2, \ldots, n\}$), and let $m = |E|$ denote the number of edges.

**Combinatorial optimization.** We consider *combinatorial optimization* (CO) problems on graphs with finite discrete *decisions* on nodes. Indeed, many CO problems can be (re-)formulated on graphs (Khalil et al., 2017). A CO problem $\mathrm{CO}(f, \mathcal{C}, d)$ is defined by an *optimization objective* $f : d^n \to \mathbb{R}_+$, *constraints* defined by a *feasible set* $\mathcal{C} \subseteq d^n$, and a set of *possible decisions* $d$ (on each $v \in V$). Given decisions $X_v \in d$ on all the nodes $v \in V$, we have a *full decision* $X \in d^n$.

For each graph $G = (V, E, W)$, we can use the optimization objective function $f$ to evaluate each full decision $X \in d^n$ on $G$ by $f(X; G)$, and we aim to *optimize* $\min_{X \in \mathcal{C}(G)} f(X; G)$.[1] By default, we consider CO problems with *binary* decisions (i.e., $d = \{0, 1\}$). Given $X \in \{0, 1\}^n$, we call each node $v$ with $X_v = 1$ a *chosen node*, and call $V_X := \{v \in V : X_v = 1\} \subseteq V$ the *chosen subset*.

### 2.2 BACKGROUND: PROBABILISTIC-METHOD-BASED UL4CO

We shall introduce the background of probabilistic-method-based unsupervised learning for combinatorial optimization (UL4CO), including the overall pipeline and some existing ideas/techniques.

#### 2.2.1 THE PROBABILISTIC-METHOD-BASED UL4CO PIPELINE: ERDŐS GOES NEURAL

The probabilistic-method-based UL4CO pipeline, Erdős Goes Neural (Karalias & Loukas, 2020), has three components: objective construction, differentiable optimization, and derandomization.

**Probabilistic objective construction.** Given a CO problem $\mathrm{CO}(f : \{0, 1\}^n \to \mathbb{R}, \mathcal{C}, d = \{0, 1\})$,[2] we first construct a *penalized* objective $f_{\mathrm{pen}}(X) = f(X) + \beta \mathbb{1}(X \notin \mathcal{C})$ with *constraint coefficient* $\beta > 0$. Then, a *probabilistic objective* $\tilde{f} : [0, 1]^n \to \mathbb{R}$ accepting probabilistic (and thus continuous) inputs is constructed such that $\tilde{f}(p) \geq \mathbb{E}_{X \sim p} f_{\mathrm{pen}}(X) = \mathbb{E}_{X \sim p} f(X) + \beta \Pr_{X \sim p}[X \notin \mathcal{C}]$.

We see each $p \in [0, 1]^n$ as a vector of probabilities, and see the $p_v$'s for $v \in [n]$ as *independent Bernoulli* variables. Hence, we have $\Pr_p[X] = \prod_{v \in V_X} p_v \prod_{u \in [n] \setminus V_X} (1 - p_u)$, $\mathbb{E}_{X \sim p} f(X) = \sum_{X \in \{0,1\}^n} \Pr_p[X] f(X)$, and $\Pr_{X \sim p}[X \notin \mathcal{C}] = \sum_{X \in \{0,1\}^n \setminus \mathcal{C}} \Pr_p[X] = 1 - \sum_{X \in \mathcal{C}} \Pr_p[X]$.

**Differentiable optimization.** For *differentiable* optimization, we need ensure that $\tilde{f}$ is differentiable (w.r.t. $p$). At this moment, let us assume it holds. Then, given each test instance $G$, we can use differentiable optimization to obtain an optimized probabilities $p_o$ with (ideally) small $\tilde{f}(p_o; G)$.

**Derandomization.** Finally, *derandomization* is used to obtain deterministic full decisions. For each test instance $G$, the derandomization process transforms each $p_o \in [0, 1]^n$ obtained by probabilistic optimization into a *discrete* full decision $X_p \in \{0, 1\}^n$. Karalias & Loukas (2020) showed a quality guarantee of derandomization by *random sampling*. See Appendix A for more details.

---

[1] We can always reverse the sign of $f$ to make it a minimization problem.

[2] Karalias & Loukas (2020) only considered binary problems. We will discuss non-binary cases in Sec 4.5.

### 2.2.2 Local Derandomization with Entry-wise Concavity

As Wang et al. (2022) pointed out, the theoretical quality guarantee by Karalias & Loukas (2020) is obtained by random sampling, and we may need a large number of samplings (and good luck) to have a good bound. Wang et al. (2022) further proposed a principle to have a *deterministic* (i.e., not relying on random sampling) quality guarantee by *iterative rounding* (i.e., a series of local derandomization along with a node enumeration). Their principle involves two concepts: (1) *local derandomization* of probabilities $p$ and (2) *entry-wise concavity* of probabilistic objective $\tilde{f}$.

**Local derandomization.** Given $p \in [0,1]^n$, $i \in [n]$, and $x \in \{0,1\}$, $\mathrm{der}(i,x;p) \in [0,1]^n$ is the result after $p_i$ being *locally derandomized* as $x$, i.e., $\mathrm{der}(i,x;p)_i = x$, and $\mathrm{der}(i,x;p)_j = p_j, \forall j \neq i$.

**Entry-wise concavity.** A probabilistic objective $\tilde{f} : [0,1] \rightarrow \mathbb{R}$ is *entry-wise concave* if $p_i \tilde{f}(\mathrm{der}(i,1;p)) + (1 - p_i)\tilde{f}(\mathrm{der}(i,0;p)) \leq \tilde{f}(p), \forall p, i$. Wang et al. (2022) showed that applying a series of local derandomization to $p$ with an entry-wise concave objective $\tilde{f}$ does not increase the objective. See Appendix A for more details and Appendix B for more extensive related work.

## 3 The Proposed Method: UCom2

Although Karalias & Loukas (2020) and Wang et al. (2022) proposed several *good properties* of probabilistic objectives, they did not propose any *practical* scheme for constructing probabilistic objectives satisfying the good properties they formalized.[3] Specifically, a practitioner would not know how to construct probabilistic objectives for new problems. Also, derandomization is still underexplored, without many discussions or techniques. This makes applying probabilistic-method-based UL4CO to specific problems difficult, especially when it involves *complex conditions* that are mathematically hard to handle. Motivated by the situation, we propose UCom2 (**U**supervised **Com**binatorial Optimization **U**nder **Com**plex Conditions), consisting of two high-level schemes:

- **(S1)** a probabilistic objective constructions scheme satisfying the good properties;
- **(S2)** a fast and effective derandomization scheme with a quality guarantee.

### 3.1 Principled Probabilistic Objectives: Expectations are All You Need

We summarize the *good* properties of a probabilistic objective $\tilde{f}$ by Karalias & Loukas (2020) and Wang et al. (2022): **(P1)** $\tilde{f} : [0,1]^n \rightarrow \mathbb{R}$ accepts *continuous* inputs $p \in [0,1]^n$ (rather than discrete $X \in \{0,1\}^n$), **(P2)** $\tilde{f}$ is an *upper bound* of the expectation of a penalized objective $f + \beta \mathbb{1}(X \notin \mathcal{C})$ for some $\beta > 0$, **(P3)** $\tilde{f}$ is *differentiable* w.r.t. $p$, and **(P4)** $\tilde{f}$ is *entry-wise concave* w.r.t. $p$.

**Target 1** (Principled objective construction). For each problem with an optimization objective $f : \{0,1\}^n \rightarrow \mathbb{R}$ and constraints $X \in \mathcal{C}$, we aim to find a *principled* way to construct a good probabilistic objective $\tilde{f} : [0,1]^n \rightarrow \mathbb{R}$ to satisfy *all the good properties* (P1)-(P4).

We shall show that *expectations are all you need*. Specifically, any probabilistic objective that can be rephrased as an expectation would satisfy properties (P1), (P3), and (P4). In other words, we only need to find an upper bound of the original (penalized) objective to satisfy (P2) too.

**Theorem 1** (Expectations are differentiable and entry-wise concave). For any $g : \{0,1\}^n \rightarrow \mathbb{R}$, $\tilde{g} : [0,1]^n \rightarrow \mathbb{R}$ with $\tilde{g}(p) = \mathbb{E}_{X \sim p} g(X)$ is differentiable and entry-wise concave w.r.t. $p$.

*Proof.* See Appendix C for all the proofs. $\square$

**Notes.** Differentiability and entry-wise concavity are closed under addition. Hence, a linear combination of expectations also works. Also, probabilities are special expectations of indicator functions. By Thm. 1, we propose the following scheme to construct *an expectation of upper bound*.

**Scheme 1** (Construct an expectation of upper bound). Given an optimization objective $f : \{0,1\}^n \rightarrow \mathbb{R}$ and constraints $X \in \mathcal{C}$, if we find $\hat{f}_1, \hat{f}_2 : \{0,1\}^n \rightarrow \mathbb{R}$ such that $\hat{f}_1(X) \geq f(X)$ and $\hat{f}_2(X) \geq \mathbb{1}(X \notin \mathcal{C}), \forall X$, then $\tilde{f}(p) := \mathbb{E}_{X \sim p}\hat{f}_1(X) + \beta \mathbb{E}_{X \sim p}\hat{f}_2(X)$ with $\beta > 0$ satisfies (P1)-(P4).

### 3.2 Incremental Greedy Derandomization: Fast and Effective

**Target 2** (Fast and effective derandomization). We aim to propose a derandomization scheme that is (1) *fast* in speed and (2) *effective* in generating high-quality discrete solutions.

---

[3]Wang et al. (2022) proposed to learn an entry-wise concave objective with the actual objective unknown.

We generalize *greedy* algorithms to greedy derandomization and propose an *incremental* scheme to improve the speed. For greedy derandomization, starting from $p_{\text{cur}} = p_{\text{o}}$, we repeat the following steps: (1) greedily finding the best local derandomization, i.e., $(i^*, x^*) \leftarrow \arg\min_{(i,x) \in [n] \times \{0,1\}} \tilde{f}(\text{der}(i, x; p_{\text{cur}}))$ and (2) conducting it, i.e., $p_{\text{cur}} \leftarrow \text{der}(i^*, x^*; p_{\text{cur}})$.

Applying greedy derandomization with an entry-wise concave $\tilde{f}$ (ensured by Scheme 1) not only has the quality guarantee (G1)-(G2) by iterative rounding, but also can find a *local minimum* (G3).

**Theorem 2** (Goodness of greedy derandomization)**.** For any entry-wise concave $\tilde{f}$ and any $p_{\text{o}} \in [0,1]^n$, the above process of greedy derandomization can always reach a point where the final $p_{\text{final}}$ is **(G1)** discrete (i.e., $p_{\text{final}} \in \{0,1\}^n$), **(G2)** no worse than $p_{\text{o}}$ (i.e., $\tilde{f}(p_{\text{final}}) \leq \tilde{f}(p_{\text{o}})$), and **(G3)** a local minimum (i.e., $\tilde{f}(p_{\text{final}}) \leq \min_{(i,x) \in [n] \times \{0,1\}} \tilde{f}(\text{der}(i, x; p_{\text{final}}))$).

Challenges naturally arise regarding the time complexity since the naive way requires $2n$ evaluations of $\tilde{f}$ at each step. We propose to conduct the derandomization in an *incremental* manner.

**Scheme 2** (Conduct incremental greedy derandomization)**.** We conduct *greedy derandomization* and improve the speed by deriving the *incremental differences* $\Delta \tilde{f}(i, x, p_{\text{cur}}) := \tilde{f}(\text{der}(i, x; p_{\text{cur}})) - \tilde{f}(p_{cur})$ for all the $(i, x)$ pairs, instead of computing the "whole" $\tilde{f}(\text{der}(i, x; p_{\text{cur}}))$'s.[4]

## 4 Applications of UCom2 to Complex Conditions

UCom2 is *general* and *practical*, i.e., can be applied to any condition with proper derivation. Below, we consider several complex conditions that are commonly involved in various CO problems (see Sec. 5 and Appendix F). For each condition, we shall derive (1) a probabilistic objective $\tilde{f}$ following Scheme 1 and (2) incremental differences of $\tilde{f}$ following Scheme 2 for derandomization. Some conditions here were encountered in existing works, but the conditions were not properly handled within the probabilistic pipeline. See the detailed discussions in Appendix A.4.

> **Template.** Below is the practical template of how we analyze each condition:
> - (S1-1) Given an optimization objective $f$, we find a good upper bound $\hat{f}(X) \geq f(X), \forall X$
> OR (S1-1′) Given a constraint $X \in \mathcal{C}$, we find a good upper bound $\hat{f}(X) \geq \mathbb{1}(X \notin \mathcal{C}), \forall X$.
>   – A good upper bound should be easy to derive and efficient to compute.
> - (S1-2) After finding $\hat{f}$, we construct $\tilde{f}(p) := \mathbb{E}_{X \sim p} \hat{f}(X)$ and derive its detailed formula.
> - (S2) We analyze the incremental differences $\tilde{f}(\text{der}(i, x; p))$.

### 4.1 Cardinality Constraints

**Definition.** We consider constraints $X \in \mathcal{C}$ with $\mathcal{C} = \{X : |V_X| \in C_c\}$. Some typical cases are $C_c = \{k\}$ or $C_c = \{t \in \mathbb{N} : t \leq k\}$ for some $k \in \mathbb{N}$ (Buchbinder et al., 2014).
Given $p \in [0,1]^n$, $|V_X| = \sum_{i \in V} X_i$ (see Sec. 2) follows a Poisson binomial distribution $\text{PoiBin}(p_1, p_2, \ldots, p_n)$ with parameters $(p_i)_{i \in [n]}$ (Wang, 1993). The probability mass function (PMF) is $\Pr_{X \sim p}[|V_X| = t] = \sum_{V_t \subseteq V : |V_t| = t} \prod_{i \in V_t} p_i \prod_{j \in V \setminus V_t} (1 - p_j)$, for each $0 \leq t \leq n$.

**Objective construction.** We (S1-1) find $\hat{f}_{\text{card}}(X; C_c) := \min_{k \in C_c} ||V_X| - k|$,[5] and (S1-2) construct $\tilde{f}_{\text{card}}(p; C_c) := \mathbb{E}_{X \sim p} \hat{f}_{\text{card}}(X; C_c)$. See the following lemma for the detailed formula of $\tilde{f}_{\text{card}}$.

**Lemma 1** ($\tilde{f}_{\text{card}}$ abides by Scheme 1)**.** For any $p \in [0,1]^n$ and $C_c$, $\tilde{f}_{\text{card}}(p; C_c) = \sum_{t \in [n] \setminus C_c} \Pr_{X \sim p}[|V_X| = t] \min_{k \in C_c} |t - k| \geq \Pr_{X \sim p}[|V_X| \notin C_c] = \mathbb{E}_{X \sim p} \mathbb{1}(X \notin \mathcal{C})$.

**Incremental differences.** We (S2) compute the incremental differences of $\tilde{f}_{\text{card}}$. The main technical difficulty lies in the incremental differences of the distribution of a Poisson binomial variable. Our derivation is based on the recursive formula of the Poisson binomial distribution.

**Lemma 2** (Incremental differences of $\tilde{f}_{\text{card}}$ for Scheme 2)**.** For any $p, i, t$, $\Pr_{X \sim p}[|V_X \setminus \{i\}| = t] = (1 - p_i)^{-1} \sum_{s=0}^{t} q_s (p_i/(p_i - 1))^{t-s} = (p_i)^{-1} \sum_{s=0}^{n-t-1} q_{t+s+1}((p_i - 1)/p_i)^s$, where $q_s := \Pr_{X \sim p}[|V_X| = s]$. Hence, $\Delta \tilde{f}_{\text{card}}(i, 0, p; C_c) := \tilde{f}_{\text{card}}(\text{der}(i, 0; p); C_c) - \tilde{f}_{\text{card}}(p; C_c) =$

---

[4]Usually, the incremental differences are simpler than the whole function, and are easily parallelizable.
[5]We can also compute $\Pr_{X \sim p}[|V_X| \notin C_c]$, but the formula in Lem. 1 practically performs better, intuitively because it distinguishes different levels of violations. See similar ideas by Pogancic et al. (2019).

$\sum_{t \in [n] \setminus C_c} (\Pr_{X \sim p}[|V_X \setminus \{i\}| = t] - \Pr_{X \sim p}[|V_X| = t]) \min_{k \in C_c} |t - k|$, and $\Delta \tilde{f}_{\text{card}}(i, 1, p; C_c) = \sum_{t \in [n] \setminus C_c} (\Pr_{X \sim p}[|V_X \setminus \{i\}| = t - 1] - \Pr_{X \sim p}[|V_X| = t]) \min_{k \in C_c} |t - k|$.

**Implementation details.** We adapt a discrete-Fourier-transform-based method (Hong, 2013; Straka, 2017) for computing the distribution. For all conditions, we make sure each $p_i \in [\epsilon, 1 - \epsilon]$ for some small $\epsilon > 0$ for numerical stability. See Appendix D.1 for more details.

## 4.2 OPTIMUM W.R.T. A SUBSET

**Definition.** We have a pairwise score function (e.g., distance) $h : V \times V \to \mathbb{R}$ and we aim to compute $f_{\text{os}}(X) := \min_{v_X \in V_X} h(i, v_X)$ for some $i \in V$ (e.g., the shortest distance to a set of points). We fix $i \in V$ in the analysis below, and let $v_1, v_2, \ldots, v_n$ be a permutation of $V = [n]$ such that $d_1 \leq d_2 \leq \cdots \leq d_n$, where $d_j = h(i, v_j), \forall j \in [n]$.

**Objective construction.** We (S1-1) find $\hat{f}_{\text{os}}(X; i, h) := \min_{v_X \in V_X} h(i, v_X)$, and (S1-2) construct $\tilde{f}_{\text{os}}(p; i, h) := \mathbb{E}_{X \sim p} \hat{f}_{\text{os}}(X; i, h)$. See the lemma below for the detailed formula of $\tilde{f}_{\text{os}}$.

**Lemma 3** ($\tilde{f}_{\text{os}}$ abides by Scheme 1). For any $p \in [0, 1]^n$, $\tilde{f}_{\text{os}}(p; i, h) = p_{v_1} d_1 + (1 - p_{v_1}) p_{v_2} d_2 + \cdots + (\prod_{j=1}^{n-1} (1 - p_{v_j})) p_{v_n} d_n = \mathbb{E}_{X \sim p} \min_{v_X \in V_X} h(i, v_X) = \mathbb{E}_{X \sim p} f_{\text{os}}(X)$.

**Incremental differences.** We (S2) compute the incremental differences of $\tilde{f}_{\text{os}}$.

**Lemma 4** (Incremental differences of $\tilde{f}_{\text{os}}$ for Scheme 2). For any $p \in [0, 1]^n$ and $j \in [n]$, we define $q_j := (\prod_{k=1}^{j-1} (1 - p_{v_k})) p_{v_j}$, the coefficient of $d_j$ in $\tilde{f}_{\text{os}}$. Then $\Delta \tilde{f}_{\text{os}}(v_j, 0, p; i, h) = -q_j d_j + \frac{p_{v_j}}{1 - p_{v_j}} \sum_{j' > j} q_{j'} d_{j'}$,[6] and $\Delta \tilde{f}_{\text{os}}(v_j, 1, p; i, h) = \sum_{j' > j} q_{j'} (d_j - d_{j'}), \forall j \in [n]$.

## 4.3 COVERED

**Definition.** We consider conditions where some $i \in V$ needs to be *covered* (i.e., at least one neighbor of $i$ is chosen). Formally, the constraints are $X \in \mathcal{C}$ with $\mathcal{C} = \{X : \{v_X \in V_X : (v_X, i) \in E\} \neq \emptyset\}$.

**Objective construction.** We (S1-1) find $\hat{f}_{\text{cv}}(X; i) := \mathbb{1}(X \notin \mathcal{C}) = \mathbb{1}(\{v_X \in V_X : (v_X, i) \in E\} = \emptyset)$, and (S1-2) construct $\tilde{f}_{\text{cv}}(p; i) := \mathbb{E}_{X \sim p} \hat{f}_{\text{cv}}(X; i)$.

**Lemma 5** ($\tilde{f}_{\text{cv}}$ abides by Scheme 1). For any $p \in [0, 1]^n$ and $i \in [n]$, $\tilde{f}_{\text{cv}}(p; i) = \prod_{v \in [n] : (v, i) \in E} (1 - p_v) = \Pr_{X \sim p}[X \notin \mathcal{C}] = \mathbb{E}_{X \sim p} \mathbb{1}(X \notin \mathcal{C})$.

**Incremental differences.** We (S2) compute the incremental differences of $\tilde{f}_{\text{cv}}$.

**Lemma 6** (Incremental differences of $\tilde{f}_{\text{cv}}$ for Scheme 2). For any $p \in [0, 1]^n$ and $i \in [n]$, if $(i, j) \notin E$, then $\Delta \tilde{f}_{\text{cv}}(j, 0, p; i) = \Delta \tilde{f}_{\text{cv}}(j, 1, p; i) = 0$; if $(i, j) \in E$, then $\Delta \tilde{f}_{\text{cv}}(j, 0, p; i) = p_j \prod_{v \in N_i, v \neq j} (p_v - 1)$ and $\Delta \tilde{f}_{\text{cv}}(j, 1, p; i) = -\tilde{f}_{\text{cv}}(p; i)$ (i.e., $\tilde{f}_{\text{cv}}(\text{der}(j, 1; p); i) = 0$).

## 4.4 CLIQUES (OR INDEPENDENT SETS)

**Definition.** We consider conditions where the chosen nodes $V_X$ should form a clique (independent sets can be analyzed similarly). Formally, the constraints are $X \in \mathcal{C}$ with $\mathcal{C} = \{X : \binom{V_X}{2} \subseteq E\}$.

**Objective construction.** We (S1-1) find $\hat{f}_{\text{cq}}(X) := |\{(u, v) \in \binom{V_X}{2} : (u, v) \notin E\}|$, and (S1-2) construct $\tilde{f}_{\text{cq}} := \mathbb{E}_{X \sim p} \hat{f}_{\text{cq}}(X)$. See the following lemma for the detailed formula of $\tilde{f}_{\text{cq}}$.

**Lemma 7** ($\tilde{f}_{\text{cq}}$ abides by Scheme 1). For any $p \in [0, 1]^n$, $\tilde{f}_{\text{cq}}(p) = \sum_{(u, v) \in \binom{V}{2} \setminus E} p_u p_v \geq \Pr_{X \sim p}[X \notin \mathcal{C}] = \mathbb{E}_{X \sim p} \mathbb{1}(X \notin \mathcal{C})$.

**Incremental differences.** We (2) compute the incremental differences of $\tilde{f}_{\text{cq}}$.

**Lemma 8** (Incremental differences of $\tilde{f}_{\text{cq}}$ for Scheme 2). For any $p \in [0, 1]^n$ and $i \in [n]$, $\Delta \tilde{f}_{\text{cq}}(i, 0, p) = -p_i \sum_{j \in [n], j \neq i, (i, j) \notin E} p_j$, and $\Delta \tilde{f}_{\text{cq}}(i, 1, p) = (1 - p_i) \sum_{j \in [n], j \neq i, (i, j) \notin E} p_j$.

---

[6]When $p_{v_j} = 1$, we need to replace $\frac{p_{v_j}}{1 - p_{v_j}} \sum_{j' > j} q_{j'} d_{j'}$ by $\sum_{j' > j} (\prod_{1 \leq i' \leq i-1, i' \neq j} (1 - p_{v_{i'}})) p_{v_i} d_{j'}$. Recall that we make sure each $p_i \in [\epsilon, 1 - \epsilon]$ for some $\epsilon > 0$, so this does not happen in practice.

### 4.5 NON-BINARY DECISIONS

**Definition.** We have been considering problems with binary decisions. We also consider *non-binary decisions*, i.e., there are (potentially) more than two decisions ($|d| \geq 2$). WLOG, we assume that $d = \{0, 1, \ldots, c-1\}$ for some $c \geq 2$. Typical examples include problems with partition or coloring.

**Notations.** For CO problems with non-binary decisions, with a slight abuse of notations, we shall extend some notations and concepts used for binary cases, while using the same symbols. With non-binary decisions $d = \{0, 1, \ldots, c-1\}$, let $X \in d^n$ denote a discrete full decision. We use $p \in [0,1]^{n \times c}$ with $\sum_{r=0}^{c-1} p_{ir} = 1, \forall i \in [n]$ to represent the probabilities of possible decisions, where each entry $p_{ir}$ represents $\Pr[X_i = r]$. Each row $p_i$ represents a categorical distribution independent of each other. Given $p \in [0,1]^{n \times c}$, $i \in [n]$, and $x \in d = \{0, 1, \ldots, c-1\}$, now $\mathrm{der}(i, x; p)$ is the result after the $i$-th row of $p$ being locally derandomized w.r.t. its $x$-th entry, i.e., $\mathrm{der}(i, x; p)_{ix} = 1, \mathrm{der}(i, y; p)_{iy} = 0, \forall y \neq x$, and $\mathrm{der}(i, x; p)_{jz} = p_{jz}, \forall j \neq i, \forall z$.

**Theoretical analysis on general non-binary cases.** Our theoretical results (Thms. 1 and 2) can be extended to non-binary cases. We also extend the theoretical results in the existing works by Karalias & Loukas (2020) and Wang et al. (2022) to non-binary cases (see Appendix E.1).

Now, for non-binary cases, a probabilistic objective $\tilde{f} : [0,1]^{n \times c} \to \mathbb{R}$ is *entry-wise concave* if $\sum_{r \in d} p_{ir} \tilde{f}(\mathrm{der}(i, r; p)) \leq \tilde{f}(p), \forall p, i$.

**Theorem 3** (Expectations are differentiable and entry-wise concave (non-binary version))**.** For any function $g : d^n \to \mathbb{R}$, $\tilde{g} : [0,1]^{n \times c} \to \mathbb{R}$ with $\tilde{g}(p) = \mathbb{E}_{X \sim p} g(X)$ is differentiable and entry-wise concave, where $\mathbb{E}_{X \sim p} g(X) = \sum_{X \in d^n} \Pr_p[X] g(X)$ with $\Pr_p[X] = \prod_{v \in [n]} p_{v X_v}$.

With non-binary decisions, the greedy derandomization is done in a similar way, as follows: (1) $(i^*, x^*) \leftarrow \arg\min_{(i,x) \in [n] \times d} \tilde{f}(\mathrm{der}(i, x; p_{\mathrm{cur}}))$ and (2) $p_{\mathrm{cur}} \leftarrow \mathrm{der}(i^*, x^*; p_{\mathrm{cur}})$.

**Theorem 4** (Goodness of greedy derandomization (non-binary version))**.** Theorem 2 still holds with $\{0, 1\}$ being replaced by any general non-binary $d$.

Due to the generality of non-binary conditions, objective construction and derandomization details are deferred to where each specific problem is analyzed in Sec. 5.

### 4.6 UNCERTAINTY

We also consider *uncertainty* in edge existence, i.e., edge probabilities $P : E \to [0, 1]$, where each edge $e \in E$ exists with probability $P(e)$. Due to the generality of uncertainty, the details of objective construction and derandomization will be deferred to where each specific problem is analyzed.

## 5 APPLICATIONS OF UCOM2 TO PROBLEMS WITH COMPLEX CONDITIONS

The conditions analyzed in Sec 4 are commonly involved in different CO problems. In this section, we apply UCOM2 to several CO problems (facility location, maximum coverage, and robust coloring) with both theoretical values (NP-hardness (Mihelic & Robic, 2004; Yanez & Ramirez, 2003)) and real-world implications. See Appendix F for the applications to four more problems (robust $k$-clique, robust dominating set, clique cover, and minimum spanning tree). For the first time, we derive probabilistic objectives for such problems, together with the derandomization process. Specifically, for each specific problem, we shall (1) check which conditions are involved and (2) construct the probabilistic objective and derandomization process based on the analyses in Sec. 4.

---

**Template.** Below is the practical template of how we analyze each problem:
- (1) Find the optimization objective ($f = \sum_i f_i$) and the constraints ($X \in \bigcap_i \mathcal{C}_i$).
- (2) Find the $\tilde{f}_i$'s for the optimization objectives and $\tilde{g}_j$'s for the the constraints.
- (3) Construct the final objective: $\sum_i \tilde{f}_i + \beta \sum_j \tilde{g}_j$ with constraint coefficient $\beta > 0$.

---

### 5.1 FACILITY LOCATION

The *facility location* problem is abstracted from real-world scenarios where we aim to find some good locations among candidate locations (Owen & Daskin, 1998; Drezner & Hamacher, 2004).

**Definition.** Given (1) a complete weighted graph $G = (V = [n], E = \binom{V}{2}, W)$, where each pair $(u, v)$ of nodes are adjacent with distance $W(u, v)$ (for the ease of presentation, we let $W(v, v) =$

$0, \forall v \in V$), and (2) the number $k$ of locations to choose, we aim to find a subset $V_X \subseteq V$ such that (c1) $|V_X| = k$, and (c2) $\sum_{v \in V} \min_{v_X \in V_X} W(v, v_X)$ is minimized.

**Involved conditions:** (1) cardinality constraints and (2) optimum w.r.t. a subset (see Secs. 4.1-4.2).

**Details.** Given $p \in [0, 1]^n$ and $\beta > 0$, $\tilde{f}_{\mathrm{FL}}(p; G, k) = (\sum_{v \in V} \tilde{f}_{\mathrm{os}}(p; v, W)) + \beta \tilde{f}_{\mathrm{card}}(p; \{k\})$. The incremental differences are $\Delta \tilde{f}_{\mathrm{FL}}(i, x, p; G, k) = \sum_{v \in V} \Delta \tilde{f}_{\mathrm{os}}(i, x, p; v, W) + \beta \Delta \tilde{f}_{\mathrm{card}}(i, x, p; \{k\}), \forall i \in [n], x \in \{0, 1\}$ See Lems. 1-4 for the details.

## 5.2 MAXIMUM COVERAGE

The *maximum coverage* problem (Khuller et al., 1999) is a classical combinatorial optimization problem with real-world applications including public traffic management (Ali & Dyo, 2017), web management (Saha & Getoor, 2009), and scheduling (Marchiori & Steenbeek, 2000).

**Definition.** Given (1) $m$ items (WLOG, $[m]$), each with weight $W_j, \forall j \in [m]$, (2) a family of $n$ sets $\mathcal{S} = \{S_1, S_2, \ldots, S_n\}$ with each $S_i \subseteq [m]$ and (3) the number $k$ of sets to choose, we aim to find a subset of sets $\mathcal{S}_X \subseteq \mathcal{S}$ such that (c1) $|\mathcal{S}_X| = k$, and (c2) the total weights of the covered items $\sum_{j \in T_X} W_j$ is maximized, where $T_X := \bigcup_{S_i \in \mathcal{S}_X} S_i$ is the set of covered items.

**Involved conditions:** (1) cardinality constraints and (2) covered (see Secs. 4.1 & 4.3).

**Details.** Construct a bipartite graph $G_{\mathcal{S}} = (V = \mathcal{S} \cup [m], E)$, where $(S_i, j) \in E$ if and only if $j \in S_i$. Given $p \in [0, 1]^n$ and $\beta > 0$, $\tilde{f}_{\mathrm{MC}}(p; \mathcal{S}, k) = \sum_{j \in [m]} W_j \tilde{f}_{\mathrm{cv}}(p; j, G_{\mathcal{S}}) + \beta \tilde{f}_{\mathrm{card}}(p; \{k\})$. The incremental differences are $\Delta \tilde{f}_{\mathrm{MC}}(i, x, p; \mathcal{S}, k) = \sum_{j \in [m]} W_j \Delta \tilde{f}_{\mathrm{cv}}(i, x, p; j, G_{\mathcal{S}}) + \beta \Delta \tilde{f}_{\mathrm{card}}(i, x, p; \{k\}), \forall i \in [n], x \in \{0, 1\}$. See Lems. 1, 2, 5, and 6 for the details.

## 5.3 ROBUST COLORING

The *robust coloring* problem (Yanez & Ramirez, 2003) generalizes the coloring problem (Jensen & Toft, 2011). It is motivated by real-world scheduling problems where some conflicts can be uncertain, with notable applications to supply chain management (Lim & Wang, 2005).

**Definition.** Given (1) an uncertain graph $G = (V, E, P)$, where $E_h := \{e \in E : P(e) = 1\}$ represents *hard conflicts* which we *must* avoid, and $E_s := \{e \in E : P(e) < 1\}$ are *soft conflicts* which possibly happen, and (2) the number $c$ of colors, we aim to find a $c$-coloring $X$ on $V$, where each node $v \in V$ has a color $X_v \in d := \{0, 1, \ldots, c - 1\}$, such that (c1) no hard conflicts are violated (i.e., $X_u \neq X_v, \forall (u, v) \in E_h$), and (c2) the probability that no violated soft conflicts happen (i.e., $\prod_{e=(u,v) \in E_s : X_u = X_v} (1 - P(e))$) is maximized. We fix $G$ and $c$ in the analysis below.

**Involved conditions:** (1) independent sets (the monochromatic subgraph of each color should be an independent set), (2) uncertainty, and (3) non-binary decisions (see Secs. 4.4-4.6).

**Objective construction.** Regarding (c1), we extend the ideas in Sec. 4.4 (Lems. 7-8) to non-binary cases. Let $\mathcal{C}_1 = \{X : (c1) \text{ is satisfied}\}$, we let $\hat{g}_1(X) := |\{X_u \neq X_v : \forall (u, v) \in E_h\}| \geq g_1(X) := \mathbb{1}(X \notin \mathcal{C}_1)$ and let $\tilde{g}_1(p) := \mathbb{E}_{X \sim p} \hat{g}_1(X)$. Regarding (c2), maximizing $\prod_{e=(u,v) \in E_s : X_u = X_v} (1 - P(e))$ is equivalent to minimizing $f_2(X) = \sum_{e=(u,v) \in E_s : X_u = X_v} -\log(1 - P(e))$. We let $\hat{f}_2(X) := f_2(X)$ and let $\tilde{f}_2(p) := \mathbb{E}_{X \sim p} \hat{f}_2(X)$. The final objective is $\tilde{f}_{\mathrm{RC}} = \tilde{f}_2 + \beta \tilde{g}_1$ with constraint coefficient $\beta > 0$. See the following lemma for the detailed formula of $\tilde{f}_{\mathrm{RC}}$.

**Lemma 9** ($\tilde{f}_{\mathrm{RC}}$ abides with Scheme 1)**.** Fix $\beta > 0$, for any $p \in [0, 1]^{n \times c}$, $\tilde{f}_{\mathrm{RC}}(p) = \sum_{e=(u,v) \in E_s} \sum_r p_{ur} p_{vr} -\log(1 - P(e)) + \beta \sum_{(u,v) \in E_h} \sum_{r=0}^{c-1} p_{ur} p_{vr} \geq \mathbb{E}_{X \sim p}(f_2(X) + \beta g_1(X))$.

**Incremental differences.** We compute the incremental differences of $\tilde{f}_{\mathrm{RC}}$.

**Lemma 10** (Incremental differences of $\tilde{f}_{\mathrm{RC}}$ for Scheme 2)**.** Fix $\beta > 0$, for any $p, i, x$, $\Delta \tilde{f}_{RC}(i, x, p) = \Delta \tilde{f}_2(i, x, p) + \beta \Delta \tilde{g}_1(i, x, p)$ with $\Delta \tilde{g}_1(i, x; p) = \sum_{x' \in d \setminus \{x\}} p_{ix'} \sum_{(i,j) \in E_h} (p_{jx} - p_{jx'})$ and $\Delta \tilde{f}_2(i, x; p) = \sum_{x' \in d \setminus \{x\}} p_{ix'} \sum_{(i,j) \in E_s} (p_{jx'} - p_{jx}) \log(1 - P(i, j))$.

## 6 EXPERIMENTS

In this section, through experiments on problems with complex conditions, we show the effectiveness of UCOM2 w.r.t. both optimization quality and time, compared to various baseline methods.

Figure 1: Trade-off plots on facility location (FL) and maximum coverage (MC). Running time: smaller the better. Objective: for FL smaller the better; for MC larger the better. For MC, we reverse the $y$-axis so that the ideal point is always at the bottom left corner.

## 6.1 FACILITY LOCATION AND MAXIMUM COVERAGE

We conduct experiments on the facility location problem and the maximum coverage problem (Secs. 5.1 & 5.2). For both problems, we mainly follow the experimental settings by Wang et al. (2023), with additional datasets and baselines. For fair comparisons with the method proposed by Wang et al. (2023), we consider inductive settings (training and test sets are different) and use the same GNN architectures.[7] See Appendix G.1 for the detailed experimental settings.

**Methods.** We compare UCOM2 with both traditional methods and ML methods: (1) **random:** $k$ locations or sets are picked uniformly at random; (2) **greedy:** deterministic greedy algorithms; (3-4) **Gurobi** (Gurobi Optimization, LLC, 2023) and **SCIP** (Bestuzheva et al., 2021; Perron & Furnon, 2023): the problems are formulated as MIPs and the two solvers are used; (5) **CardNN** (Wang et al., 2023): a SOTA UL4CO method with three variants; (6) **CardNN-noTTO**: CardNN directly optimizes on each test graph in test time, and these are variants of CardNN without test-time optimization; (7) **EGN-naïve**: EGN (Karalias & Loukas, 2020) with a naïve probabilistic objective construction and iterative rounding; (8) **RL**: a reinforcement-learning method (Kool et al., 2019).[8]

**Datasets.** We consider both random synthetic graphs and real-world graphs:

- **random graphs:** The number after "rand" represents the size of the random graphs in the group. Each group of random graphs contains 100 graphs generated from the same distribution.
- **real-world graphs**: For facility location, each graph contains real-world entities with locations (*starbucks*, *mcd*, *subway*). For maximum coverage, each graph contains real-world sets (*twitch*, *railway*). Each group of real-world graphs also contains multiple graphs from the same source.

**Speed-quality trade-offs.** Several methods allow speed-quality trade-offs. That is, we can grant more running time to obtain better optimization quality. For UCOM2, we use test-time augmentation (Jin et al., 2023) on the test graphs by adding perturbations into both graph topology and features to obtain additional data. The three variants of UCOM2 are obtained by using different numbers of additional augmented data and taking the best objective.

**Results.** We show the results in Tabs. 1 & 2.[9] For each group of datasets and each method, we show the (normalized) optimization objective and the running time. For each group of datasets, the performance is averaged on all the graphs in the group. Averaged over all the groups of datasets, we further compute the average objective, time, and ranks. The average rank "sum" (ARS) is the summation of the average ranks w.r.t. objective and time. The proposed method UCOM2 achieves the best trade-offs overall. On facility location, the top-3 methods w.r.t. ARS are the three variants of UCOM2. On maximum coverage, the three variants are ranked 1, 3, and 4 w.r.t. ARS, respectively. The comparisons between UCOM2 and EGN-naive show the empirical effectiveness of our two schemes, which are the differences between UCOM2 and EGN-naive. In Fig. 1, we show the detailed trade-offs on the random graphs, which visually supports that our method achieves the best trade-off overall. See Appendix G for the full results and the ablation study.

## 6.2 ROBUST COLORING

We conduct experiments on the robust coloring problem (see Sec. 5.3) under transductive settings (i.e., directly optimize probabilistic decisions $p$ on each test dataset). See Appendix G.1 for details.

**Methods.** We compare UCOM2 with four baseline methods: (1-2) **greedy-RD** and **greedy-GA**: both methods decide the colors following an enumeration of nodes, where greedy-RD follows a ran-

---

[7]See Appendix H.1 for discussions on inductive settings and transductive settings.

[8]See Appendix H.2 for discussions on reinforcement learning and probabilistic-method-based UL4CO.

[9]The results are averaged over random trials, see Appendix G for the full results with standard deviations.

Table 1: Results on facility location. Running time (in seconds): smaller the better. Objective (obj): smaller the better. In each column, ▉ indicates ranking 1st, ▉ ranking 2nd, and ▉ ranking 3rd.

| Method | rand500 obj↓ | rand500 time↓ | rand800 obj↓ | rand800 time↓ | starbucks obj↓ | starbucks time↓ | mcd obj↓ | mcd time↓ | subway obj↓ | subway time↓ | average obj↓ | average time↓ | avg rank obj↓ | avg rank time↓ | avg rank sum↓ |
|---|---|---|---|---|---|---|---|---|---|---|---|---|---|---|---|
| random | 1.41 | 642.34 | 1.51 | 140.99 | 1.87 | 187.12 | 1.69 | 119.65 | 1.57 | 111.62 | 1.61 | 240.34 | 12.0 | 13.6 | 25.6 |
| greedy | 1.18 | 5.63 | 1.16 | 3.45 | 1.22 | 5.07 | 1.18 | 5.74 | 1.11 | 12.09 | 1.17 | 6.40 | 8.4 | 3.8 | 12.2 |
| Gurobi | 1.08 | 325.60 | 1.22 | 72.91 | 1.07 | 80.17 | 1.37 | 62.12 | 2.58 | 61.09 | 1.46 | 120.38 | 9.4 | 11.4 | 20.8 |
| SCIP | 1.75 | 330.46 | 2.35 | 110.34 | 24.42 | 64.11 | 54.49 | 240.20 | 55.90 | 334.37 | 27.78 | 215.90 | 15.8 | 12.8 | 28.6 |
| CardNN-S | 1.14 | 34.96 | 1.06 | 7.74 | 1.81 | 11.85 | 1.96 | 1.00 | 1.09 | 10.04 | 1.41 | 13.12 | 9.2 | 3.8 | 13.0 |
| CardNN-GS | 1.00 | 198.07 | 1.01 | 83.97 | 1.48 | 54.43 | 1.14 | 19.92 | 1.13 | 14.60 | 1.15 | 74.20 | 6.0 | 8.4 | 14.4 |
| CardNN-HGS | 1.00 | 322.33 | 1.01 | 118.05 | 1.06 | 76.51 | 1.11 | 45.55 | 1.03 | 26.18 | 1.04 | 117.73 | 3.4 | 11.0 | 14.4 |
| CardNN-noTTO-S | 1.58 | 5.19 | 1.34 | 1.26 | 4.95 | 1.50 | 1.14 | 9.13 | 2.54 | 1.00 | 2.31 | 3.62 | 11.8 | 2.4 | 14.2 |
| CardNN-noTTO-GS | 1.14 | 76.99 | 1.15 | 30.83 | 1.44 | 25.14 | 1.12 | 31.23 | 1.14 | 3.64 | 1.20 | 33.56 | 8.0 | 6.2 | 14.2 |
| CardNN-noTTO-HGS | 1.14 | 114.96 | 1.01 | 118.05 | 1.58 | 21.22 | 1.25 | 14.03 | 1.12 | 21.27 | 1.22 | 57.91 | 7.6 | 8.0 | 15.6 |
| EGN-naïve | 1.10 | 210.90 | 1.14 | 50.11 | 1.15 | 94.24 | 1.64 | 23.97 | 1.47 | 56.22 | 1.30 | 87.09 | 8.4 | 10.2 | 18.6 |
| RL-transductive | 2.31 | 802.92 | 2.25 | 176.23 | 10.22 | 1403.43 | 2.73 | 897.34 | 2.52 | 837.17 | 4.01 | 823.42 | 14.8 | 15.6 | 30.4 |
| RL-inductive | 1.69 | 803.07 | 1.85 | 176.55 | 2.74 | 233.93 | 2.52 | 149.57 | 2.37 | 139.55 | 2.23 | 300.54 | 13.4 | 15.0 | 28.4 |
| UCom2-short | 1.05 | 1.00 | 1.04 | 1.00 | 1.06 | 1.00 | 1.10 | 1.78 | 1.06 | 5.23 | 1.06 | 2.00 | 4.4 | 1.6 | 6.0 |
| UCom2-middle | 1.00 | 83.35 | 1.00 | 27.89 | 1.01 | 1.99 | 1.03 | 5.05 | 1.05 | 13.03 | 1.02 | 26.26 | 2.2 | 4.6 | 6.8 |
| UCom2-long | 1.00 | 166.70 | 1.00 | 47.88 | 1.00 | 7.89 | 1.00 | 21.10 | 1.00 | 23.45 | 1.00 | 53.41 | 1.0 | 7.4 | 8.4 |

Table 2: Results on maximum coverage. Running time (in seconds): smaller the better. Objective (obj): larger the better. In each column, ▉ indicates ranking 1st, ▉ ranking 2nd, and ▉ ranking 3rd.

| Method | rand500 obj↑ | rand500 time↓ | rand1000 obj↑ | rand1000 time↓ | twitch obj↑ | twitch time↓ | railway obj↑ | railway time↓ | average obj↑ | average time↓ | avg rank obj↑ | avg rank time↓ | avg rank sum↓ |
|---|---|---|---|---|---|---|---|---|---|---|---|---|---|
| random | 0.81 | 2714.15 | 0.79 | 727.67 | 0.51 | 391.45 | 0.96 | 316.68 | 0.77 | 1037.49 | 12.5 | 14.0 | 26.5 |
| greedy | 0.98 | 1.00 | 0.99 | 1.00 | 1.00 | 1.12 | 1.00 | 1.03 | 0.99 | 1.03 | 7.3 | 1.3 | 8.5 |
| Gurobi | 1.00 | 1357.92 | 0.99 | 364.03 | 1.00 | 1.00 | 1.00 | 159.16 | 1.00 | 470.53 | 4.3 | 9.3 | 13.5 |
| SCIP | 0.97 | 1357.70 | 0.96 | 362.32 | 1.00 | 5.54 | 0.99 | 159.82 | 0.98 | 471.35 | 6.3 | 10.8 | 17.0 |
| CardNN-S | 0.93 | 131.16 | 0.92 | 35.07 | 1.00 | 13.22 | 0.97 | 4.08 | 0.96 | 45.89 | 8.0 | 5.5 | 13.5 |
| CardNN-GS | 0.99 | 439.38 | 0.99 | 208.36 | 1.00 | 27.02 | 1.00 | 21.79 | 1.00 | 174.14 | 3.0 | 9.0 | 12.0 |
| CardNN-HGS | 0.99 | 614.96 | 0.99 | 268.37 | 1.00 | 55.39 | 1.00 | 41.51 | 1.00 | 245.06 | 3.8 | 10.5 | 14.3 |
| CardNN-noTTO-S | 0.73 | 23.33 | 0.70 | 5.08 | 0.00 | 1.75 | 0.94 | 1.98 | 0.59 | 8.03 | 16.0 | 2.5 | 18.5 |
| CardNN-noTTO-GS | 0.83 | 107.76 | 0.79 | 58.17 | 0.05 | 3.14 | 0.96 | 7.82 | 0.65 | 44.22 | 13.5 | 5.0 | 18.5 |
| CardNN-noTTO-HGS | 0.83 | 144.58 | 0.79 | 74.58 | 0.05 | 4.65 | 0.96 | 8.31 | 0.66 | 58.03 | 12.0 | 6.3 | 18.3 |
| EGN-naïve | 0.92 | 1365.17 | 0.91 | 306.92 | 0.46 | 196.90 | 0.95 | 159.33 | 0.81 | 507.08 | 12.3 | 12.3 | 24.5 |
| RL-transductive | 0.92 | 3392.69 | 0.82 | 909.58 | 0.95 | 2935.89 | 0.96 | 2375.04 | 0.91 | 2403.30 | 15.5 | 15.5 | 26.8 |
| RL-inductive | 0.77 | 3393.34 | 0.77 | 910.13 | 0.59 | 492.43 | 0.96 | 398.19 | 0.77 | 1298.52 | 13.5 | 15.5 | 29.0 |
| UCom2-short | 0.99 | 10.26 | 0.99 | 7.52 | 1.00 | 2.98 | 1.00 | 2.51 | 1.00 | 5.81 | 4.5 | 3.0 | 7.5 |
| UCom2-middle | 1.00 | 166.14 | 1.00 | 25.82 | 1.00 | 18.65 | 1.00 | 9.66 | 1.00 | 55.07 | 3.3 | 6.8 | 10.0 |
| UCom2-long | 1.00 | 330.35 | 1.00 | 236.95 | 1.00 | 31.57 | 1.00 | 18.76 | 1.00 | 154.41 | 2.3 | 9.0 | 11.3 |

Table 3: Results on robust coloring. Running time (in seconds): smaller the better. Objective: smaller the better. In each column, ▉ indicates ranking 1st, and ▉ ranking 2nd.

| Method | collins, 18 colors obj | collins, 18 colors time | collins, 25 colors obj | collins, 25 colors time | gavin, 8 colors obj | gavin, 8 colors time | gavin, 15 colors obj | gavin, 15 colors time | krogan, 8 colors obj | krogan, 8 colors time | krogan, 15 colors obj | krogan, 15 colors time | ppi, 47 colors obj | ppi, 47 colors time | ppi, 50 colors obj | ppi, 50 colors time | avg rank obj | avg rank time | avg rank sum |
|---|---|---|---|---|---|---|---|---|---|---|---|---|---|---|---|---|---|---|---|
| greedy-RD | 115.33 | 300.34 | 23.42 | 300.79 | 66.51 | 300.53 | 7.36 | 301.46 | 117.47 | 300.06 | 0.87 | 301.24 | 4.16 | 301.31 | 1.23 | 301.24 | 2.88 | 3.25 | 6.13 |
| greedy-GA | 114.36 | 188.21 | 22.20 | 243.93 | 66.51 | 398.90 | 7.36 | 540.62 | 117.47 | 941.35 | 0.87 | 1256.66 | 3.66 | 1416.38 | 1.23 | 1484.27 | 2.50 | 4.25 | 6.75 |
| DC | 586.56 | 300.28 | 159.15 | 300.38 | 311.91 | 300.11 | 58.10 | 300.12 | 1065.52 | 300.07 | 1.76 | 300.46 | 43.35 | 300.13 | 6.72 | 300.76 | 5.00 | 2.50 | 7.50 |
| Gurobi | 87.28 | 301.71 | 16.23 | 306.10 | 42.41 | 300.80 | 7.28 | 303.50 | 46.78 | 300.80 | 0.87 | 51.70 | 4.60 | 328.48 | 1.31 | 313.23 | 2.50 | 4.00 | 6.50 |
| UCom2 (CPU) |  | 79.36 |  | 54.37 |  | 152.20 |  | 260.90 |  | 211.43 |  | 8.55 |  | 116.54 |  | 120.56 | 1.50 | 1.00 | 2.50 |
| UCom2 (GPU) | 82.26 | 7.09 | 15.16 | 8.03 | 42.99 | 13.28 | 6.72 | 17.25 | 53.44 | 13.73 | 0.87 | 1.91 | 2.93 | 5.24 | 1.01 | 5.48 |  |  |  |

dom (RD) permutation of the nodes while greedy-GA uses a genetic algorithm (GA) to learn the permutation;[10] (3) **Deterministic coloring (DC)**: a deterministic greedy coloring algorithm (Kosowski & Manuszewski, 2004) is used to avoid all the hard conflicts, and it tries to avoid as many soft conflicts as possible. (4) **Gurobi**: the problem is formulated as an MIP and the solver is used.

**Datasets.** We use four real-world uncertain graphs: (1) **collins**, (2) **gavin**, (3) **krogan**, and (4) **PPI**.

**Speed-quality trade-offs.** We record the running time of our method using only CPUs and using GPUs. For our method, we start from multiple random initial probabilities, while making sure that even with only CPUs, our method uses less time than each baseline.

**Results.** In Tab. 3, we show the results. For each group of datasets and each method, we show the optimization objective and the running time (without normalization). The average ranks are computed in the same way as in Tabs. 1 & 2. With the least running time, UCom2 consistently outperforms the two **greedy** baselines and **DC**, and outperforms **Gurobi** in most cases. This superiority holds even when we only use CPUs for UCom2. Moreover, when using GPUs, UCom2 is even faster.

## 7 CONCLUSION

In this work, we study unsupervised combinatorial optimization under complex conditions. We propose UCom2, which consists of a *principled* probabilistic objective construction scheme (Sec. 3.1) and a *fast* and *effective* derandomization scheme (Sec. 3.2). We provide theoretical results for the proposed schemes and further extend them to non-binary cases (Thms. 1-4). We show that UCom2 can be effectively applied to various complex conditions and problems (Secs. 4 & 5), evidenced by the empirical superiority of UCom2 in the experiments (Sec. 6).

---

[10]Greedy-GA is the method proposed by Yanez & Ramirez (2003) in the original paper of robust coloring.

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

# A  ADDITIONAL DETAILS ON THE BACKGROUND

We would like to provide some additional details on the background (Section 2.2), especially regarding the works by Karalias & Loukas (2020) and Wang et al. (2022). We would also talk about the work by Wang et al. (2023).

## A.1  CLARIFICATION: HOW IS UCOM2 DIFFERENT? ARE THESE WORKS COMPARED WITH OUR METHOD UCOM2 IN THE EXPERIMENTS?

The clearest difference between UCOM2 and the works by Karalias & Loukas (2020) and Wang et al. (2022) is the proposed derandomization scheme. Regarding probabilistic objective construction, we would like to clarify that we do not modify the pipeline proposed by Karalias & Loukas (2020), but improve the components within the pipeline. For the problems with complex conditions that we consider in the paper, both Karalias & Loukas (2020) and Wang et al. (2022) did not provide any guidelines on how to handle the conditions. Specifically, Wang et al. (2022) proposed to learn a probabilistic objective with good properties when the ground-truth objective is unknown, but did not try to construct a good probabilistic objective out of an existing objective. The main contribution of our probabilistic objective construction scheme is that the scheme gives practical guidelines to derive detailed formulae (what we did in Sections 4-5).

If the probabilistic objectives are well constructed (specifically, if the constructed objectives are entry-wise concave), the methods by Karalias & Loukas (2020) and Wang et al. (2022) are essentially the same. See the response of Wang et al. (2022).[11] The method EGN (with naive objective construction and iterative-rounding derandomization) by Karalias & Loukas (2020) is compared in our experiments. A variant of UCOM2 using iterative rounding can be seen as the method by Wang et al. (2022). Indeed, we compare UCOM2 with such a variant in our ablation study, where UCOM2 clearly outperforms the variant. See Appendix G.3.

Regarding the work by Wang et al. (2023), the authors specifically considered cardinality constraints. However, they did not follow the EGN pipeline but used optimal transport to derive a different top-$k$ technique. See Appendix A.4 for more details.

## A.2  ON THE "DIFFERENTIABLE OPTIMIZATION" IN THE PIPELINE (SECTION 2.2.1)

One can directly optimize a probabilistic decision $p$ on each test instance $G_{\text{test}}$, i.e., aim to find $p^* \approx \arg\min_p \tilde{f}(p; G_{\text{test}})$. One can also train an encoder (e.g., a graph neural network) parameterized by parameters $\theta$ on a training set $\mathcal{D}_{\text{train}}$ to learn to output "good" (probabilistic) decisions for each training instance, i.e., aim to find $\theta^* \approx \arg\min_\theta \sum_{G \in \mathcal{D}_{\text{train}}} \tilde{f}(\text{ENCODER}(G; \theta); G)$. Such a trained encoder can be applied to each test instance $G_{\text{test}}$ and output a (probabilistic) decision $p = \text{ENCODER}(G_{\text{test}}; \theta)$. Training such an encoder is optional, but if trained well, it can save time for unseen cases since we do not need to optimize $p$ for each test instance from scratch.[12] Even when using such an encoder, one can still further directly optimize the probabilistic decisions on each test instance. See more discussions on inductive settings and transductive settings in Appendix H.1.

## A.3  FORMAL THEORETICAL RESULTS IN THE EXISTING WORKS

Here, we would like to provide the detailed formal theoretical results in the existing works by Karalias & Loukas (2020) and Wang et al. (2022). Recall that Karalias & Loukas (2020) showed a quality guarantee by *random sampling*.

**Theorem 5** (Theorem 1 by Karalias & Loukas (2020)). Assume that $f$ is non-negative.[13] Fix any $\beta > \max_{X \in \mathcal{C}} f(X; G)$, $\epsilon > 0$, and $t \in (0, 1]$ such that $(1 - t)\epsilon < \beta$. If $\tilde{f}(p_{\text{init}}; G) < \beta$, then $\Pr_{X \sim p_{\text{init}}}[f(X; G) < \epsilon \wedge X \in \mathcal{C}] \geq t$.

Recall that Wang et al. (2022) further proposed a principle to guarantee the quality by *iterative rounding*. Also, recall the following definitions: given a probability decision $p \in [0, 1]^n$, an index

---

[11]https://openreview.net/forum?id=HjNn9oD_v47&noteId=lglGd6uJRvl
[12]See some related discussions at https://github.com/Stalence/erdos_neu.
[13]We can always ensure this for any bounded $f$ by adding a sufficiently large positive constant to $f$.

$i \in [n]$, and $x \in \{0, 1\}$, let $\mathrm{der}(i, x; p)$ denoted the result after the $i$-th entry of $p$ being *locally derandomized* as $x$. Formally, $\mathrm{der}(i, x; p)_i = x$, and $\mathrm{der}(i, x; p)_j = p_j, \forall j \neq i$. A probabilistic objective $\tilde{f}$ is *entry-wise concave* if $p_i \tilde{f}(\mathrm{der}(i, 1; p); G) + (1 - p_i) \tilde{f}(\mathrm{der}(i, 0; p); G) \leq \tilde{f}(p; G), \forall G, p, i$.

**Theorem 6** (Theorem 1 by Wang et al. (2022)). *If $\tilde{f}(p) \geq \mathbb{E}_{X \sim p} f(X) + \beta \Pr_{X \sim p}[X \notin \mathcal{C}], \forall p$ is entry-wise concave and non-negative with $\beta > \max(\tilde{f}(p_{\mathrm{init}}), \max_{X \in \mathcal{C}} f(X))$, then for any permutation $\pi : [n] \to [n]$, starting from $p_{\mathrm{cur}} = p_{\mathrm{init}}$ and for $i \in [n]$ doing (1) $x^* \leftarrow \arg\min_{x \in \{0,1\}} \tilde{f}(\mathrm{der}(\pi(i), x; p_{\mathrm{cur}}))$ and (2) $p_{\mathrm{cur}} \leftarrow \mathrm{der}(i, x^*; p_{\mathrm{cur}})$ will finally give a discrete $p_{\mathrm{final}} \in \mathcal{C}$ such that $f(p_{\mathrm{final}}) < \tilde{f}(p_{\mathrm{init}})$.*

### A.4 COMPLEX CONDITIONS IN EXISTING WORKS

As mentioned in Section 4, several conditions have been encountered in existing works. Here, for each condition analyzed in Section 4, we shall discuss how the existing works try to handle it and how our derivation is different and satisfies more good properties (see Section 3).

**Cardinality constraints.** Wang et al. (2023) specifically considered cardinality constraints. However, they used optimal transport soft top-$k$ instead of the probabilistic pipeline we focus on in this work. Also, our derivation is more general since it can handle general cardinality constraints other than choosing a specific number of entities (i.e., top-$k$). Wang et al. (2023) claimed that cardinality constraints cannot be handled in the EGN pipeline, but this work shows that cardinality constraints can actually be properly handled following our schemes.

**Optimum w.r.t. a subset** Wang et al. (2023) also encountered such a condition in the facility location problem which they considered. They used the $\mathrm{softmin}$ to approximate the $\min$ operation, which indeed provides an upper bound. However, the result of $\mathrm{softmin}$ is not entry-wise concave, and thus fails to satisfy the good property required by Wang et al. (2022), while our derivation satisfies all the good properties, as shown in Theorem 1.

**Covered** Wang et al. (2023) also encountered such a condition in the maximum coverage problem which they considered. They used $\min(1, \sum_{v \in N_i} p_v)$ as an approximation for the probability of $i$ being covered, where $N_i = \{v : (v, i) \in E\}$. In other words, they used $\max(0, 1 - \sum_{v \in N_i} p_v)$ to approximate the probability that $i$ is not covered. As we have shown, the probability that $i$ is not covered is exactly $\prod_{v \in N_i}(1 - p_v)$. However, $\max(0, 1 - \sum_{v \in N_i} p_v)$ is not an *upper bound* of $\prod_{v \in N_i}(1 - p_v)$ but a *lower bound*. Therefore, the derivation by Wang et al. (2023) does not satisfy the conditions required for the probabilistic pipeline (see Section 2.2.1).

**Cliques (or independent sets).** Karalias & Loukas (2020) considered the maximum clique problem, and their derivation coincides with ours. Our probabilistic objective construction scheme provides a principled way to interpret the derivation.

## B ADDITIONAL RELATED WORK

In the main text, we introduce the background with the most related works (Section 2.2). Here, we provide a more extensive literature review.

**Unsupervised learning for combinatorial optimization.** Most related are the works on unsupervised learning for combinatorial optimization (UL4CO). Karalias & Loukas (2020) first explicitly applied the probabilistic method to combinatorial optimization (CO) problems. UL4CO has been further explored and improved. Wang et al. (2022) proposed some theoretically desirable properties for probabilistic objective formulation. However, they did not propose any practical scheme to construct objectives within the probabilistic-method framework. Wang et al. (2023) focused on cardinality constraints, and instead of using the probabilistic method, they proposed a differentiable soft top-$k$ method to handle cardinality constraints. We also see the potential future direction of extending UL4CO assuming dependence between decisions instead of assuming the decisions on the node are independent (Sanokowski et al., 2023). See Section 2.2 and Appendix A for more details.

**Reinforcement learning for combinatorial optimization.** There are also other learning-based methods proposed for CO problems. Typical techniques include reinforcement learning (RL). The pioneers who applied RL to CO problems include Bello et al. (2016) and Khalil et al. (2017). Most

reinforcement-learning-for-combinatorial-optimization (RL4CO) methods focus on routing problems such as the traveling salesman problem (TSP) and the vehicle routing problem (VRP) (Berto et al., 2023; Kool et al., 2019; Kim et al., 2021; Delarue et al., 2020; Kim et al., 2022; Nazari et al., 2018; Ye et al., 2023a; Chalumeau et al., 2023; Luo et al., 2023; Grinsztajn et al., 2023), as well as maximum independent sets (MIP) (Ahn et al., 2020; Qiu et al., 2022; Sun & Yiming, 2023; Li et al., 2023), while our method and schemes are more general.

See also some recent surveys on RL4CO (Mazyavkina et al., 2021; Bengio et al., 2021; Cappart et al., 2023; Munikoti et al., 2023) for more details.

As pointed out by Wang et al. (2023) and as shown in our experimental results, the existing RL-based methods still suffer from efficiency issues. See the discussions by Wang et al. (2022). See also Appendix H.2 for some discussions on RL and probabilistic-method-based UL4CO. We also see the potential that probabilistic-method-based objective construction can also be used for reward design in RL methods.

**Other machine-learning techniques for combinatorial optimization.** Except for RL, there are also some other machine-learning techniques proposed for CO problems. We have recent progress based on search (Choo et al., 2022; Son et al., 2023; Li et al., 2023), sampling (Sun et al., 2023), graph-based diffusion (Sun & Yiming, 2023), generative flow networks (Zhang et al., 2023), meta-learning (Qiu et al., 2022; Wang & Li, 2023), and quantum machine learning (Ye et al., 2023b). Physics-inspired machine learning has also been considered (Schuetz et al., 2022a; Aramon et al., 2019; Schuetz et al., 2022b). There is also a line of research on perturbation-based methods for CO (Pogancic et al., 2019; Berthet et al., 2020; Paulus et al., 2021; Ferber et al., 2023). As shown by Wang et al. (2023), perturbation-based methods are consistently outperformed by CardNN proposed by Wang et al. (2023), while our proposed method outperforms CardNN in most cases.

## C  PROOFS

Here, we provide proof for each theoretical statement in the main text.

*Proof of Theorem 1.* For any $p$ and $i$, we have

$$\tilde{g}(p)$$
$$=\mathbb{E}_{X\sim p}g(X)$$
$$=\sum_{X\in\{0,1\}^n}\Pr_p[X]g(X)$$
$$=\sum_X\prod_{v\in V_X}p_v\prod_{u\in[n]\backslash V_X}(1-p_u)g(X)$$
$$=\sum_{X\,:\,i\in V_X}\left(\prod_{v\in V_X,v\neq i}p_v\prod_{u\in[n]\backslash V_X}(1-p_u)\right)p_ig(X)+\sum_{X\,:\,i\notin V_X}\left(\prod_{v\in V_X}p_v\prod_{u\in[n]\backslash V_X,u\neq i}(1-p_u)\right)(1-p_i)g(X)$$
$$=p_i\sum_{X\,:\,i\in V_X}\prod_{v\in V_X,v\neq i}p_v\prod_{u\in[n]\backslash V_X}(1-p_u)g(X)+(1-p_i)\sum_{X\,:\,i\notin V_X}\prod_{v\in V_X}p_v\prod_{u\in[n]\backslash V_X,u\neq i}(1-p_u)g(X)$$
$$=p_i\tilde{g}(\mathrm{der}(i,1;p))+(1-p_i)\tilde{g}(\mathrm{der}(i,0;p))$$
$$\geq p_i\tilde{g}(\mathrm{der}(i,1;p))+(1-p_i)\tilde{g}(\mathrm{der}(i,0;p)),$$

completing the proof on entry-wise concavity. Regarding differentiability, since

$$\mathbb{E}_{X\sim p}g(X)=\sum_{X\in\{0,1\}^n}\Pr_p[X]g(X),$$

it suffices to show that

$$\Pr_p[X]g(X)=\prod_{v\in V_X}p_v\prod_{u\in[n]\backslash V_X}(1-p_u)g(X)$$

is differentiable w.r.t $p$ for each $X\in\{0,1\}^n$. Indeed, fix any $X$, $\prod_{v\in V_X}p_v\prod_{u\in[n]\backslash V_X}(1-p_u)g(X)$ is a polynomial of $p_i$'s, and is thus differentiable w.r.t. $p$. $\qquad\square$

*Proof of Theorem 2.* First, we claim that for any non-discrete $p_{\text{cur}} \notin \{0,1\}^n$, we can always derandomize it through a series of local derandomization while the value of $\tilde{f}$ does not increase. This is guaranteed by the entry-wise concavity of $\tilde{f}$. Specifically, since

$$p_i \tilde{f}(\text{der}(i,1;p)) + (1-p_i)\tilde{f}(\text{der}(i,0;p)) \le \tilde{f}(p), \forall p, i,$$

we have

$$\min(\text{der}(i,1;p), \text{der}(i,0;p)) \le \tilde{f}(p), \forall p, i,$$

which implies that we can always derandomize a non-discrete entry without increasing the value of $\tilde{f}$. Therefore, if we greedily improve $\tilde{f}$ via local derandomization, we can always terminate at a discrete point, completing the proof for point (1). Point (2) is trivial since at each step we make sure that the value of $\tilde{f}$ does not increase. Point (3) is also immediate from the way we conduct local derandomization. Specifically, if the current $p_{\text{cur}}$ is not a local minimum, we can always find a possible local derandomization step to proceed the process while strictly decreasing the value of $\tilde{f}$. □

*Proof of Lemma 1.* We first validate that $\hat{f}_{\text{card}}(X) \ge \mathbb{1}(X \notin \mathcal{C}), \forall X$. Indeed, $\min_{k \in C_c} ||V_X| - k| = 0$ if $V_X \in C_c$ and $\min_{k \in C_c} ||V_X| - k| \ge 1$ otherwise. Then we validate the detailed formula of $\hat{f}_{\text{card}}$. Indeed,

$$\mathbb{E}_{X \sim p} \hat{f}_{\text{card}}(X)$$
$$= \sum_X \Pr[X] \min_{k \in C_c} ||V_x| - k|$$
$$= \sum_t \Pr_p[|V_X| = t] \min_{k \in C_c} |t - k|$$
$$= \sum_{t \in [n] \setminus C_c} \Pr_p[|V_X| = t] \min_{k \in C_c} |t - k|.$$

□

Fix any $i \in [n]$ and any $p \in [0,1]^n$, we have

$$\begin{cases} \Pr_{X \sim p}[|V_X| = 0] = \Pr_{X \sim p}[|V_X \setminus \{i\}| = 0](1 - p_i) \\ \Pr_{X \sim p}[|V_X| = t] = \Pr_{X \sim p}[|V_X \setminus \{i\}| = t](1 - p_i) + \Pr_{X \sim p}[|V_X \setminus \{i\}| = t - 1]p_i, \forall t \\ \Pr_{X \sim p}[|V_X| = n] = \Pr_{X \sim p}[|V_X \setminus \{i\}| = n - 1]p_i \end{cases} . \quad (1)$$

*Proof of Lemma 2.* Let $q_s$ denote $\Pr_{X \sim p}[|V_X| = s]$ for each $s$ as in the statement, and also let $\tilde{q}_s$ denote $\Pr_{X \sim p}[|V_X \setminus \{i\}| = s]$. By Equation (1), if we start from $q_0 = \tilde{q}_0(1 - p_i)$, we have

$$\tilde{q}_0 = \frac{q_0}{1 - p_i}, \tilde{q}_1 = \frac{q_1 - p_i \tilde{q}_0}{1 - p_i} = \frac{q_1(1 - p_i) - q_0 p_i}{(1 - p_i)^2}, \cdots,$$

which satisfies $\tilde{q}_t = (1 - p_i)^{-1} \sum_{s=0}^t q_s \left(\frac{p_i}{p_i - 1}\right)^{t-s}$. Now, if

$$\tilde{q}_t = (1 - p_i)^{-1} \sum_{s=0}^t q_s \left(\frac{p_i}{p_i - 1}\right)^{t-s}$$

holds for all $t \le T - 1$, we aim to show that it also holds for $t = T$, which shall prove the statement by mathematical induction. Indeed, we have

$$\tilde{q}_T = \frac{q_T - p_i \tilde{q}_{T-1}}{1 - p_i} = \frac{q_T - p_i(1 - p_i)^{-1} \sum_{s=0}^{T-1} q_s \left(\frac{p_i}{p_i - 1}\right)^{T-1-s}}{1 - p_i} = (1 - p_i)^{-1} \sum_{s=0}^T q_s \left(\frac{p_i}{p_i - 1}\right)^{T-s},$$

completing the proof. If we start from $q_n = \tilde{q}_{n-1} p_i$, we can obtain the another term (i.e., $(p_i)^{-1} \sum_{s=0}^{n-t-1} q_{t+s+1} \left(\frac{p_i - 1}{p_i}\right)^s$) in the statement in a similar way. □

*Proof of Lemma 3.* The formula of the expectation $\mathbb{E}_{X \sim p}[\min_{v_X \in V_X} h(i, v_x)]$ is expanded with the idea that we can consider all the nodes in $V_X$ in the ascending order w.r.t the value of $h$. With the notations in the statement, if $v_1 \in V_X$, $\min_{v_X \in V_X} h(i, v_x)$ would be $d_1$, which happens with probability $p_{v_1}$; if $v_1 \notin V_X$ but $v_2 \in V_X$, then $\min_{v_X \in V_X} h(i, v_x)$ would be $d_2$, which happens with probability $(1 - p_{v_1})p_{v_2}$. All the cases can be analyzed in a similar way, which eventually gives the formula in the statement. □

*Proof of Lemma 4.* When $p' = \mathrm{der}(v_j, 0; p)$, we have

$$\tilde{f}_{\mathrm{os}}(p'; i, h)$$

$$= p'_{v_1} d_1 + (1 - p'_{v_1})p'_{v_2} d_2 + \cdots + \left( \prod_{s=1}^{n-1}(1 - p'_{v_s}) \right) p'_{v_n} d_n$$

$$= \sum_{s<j} \prod_{k=1}^{s-1}(1 - p_{v_k})p_{v_s} d_s + 0 + \sum_{t>j} \prod_{1 \le k \le t-1, k \ne j}(1 - p_{v_k})p_{v_t} d_t$$

$$= \sum_{s<j} q_s d_s + 0 + \sum_{j'>j} \frac{1}{1 - p_{v_j}} q_{j'} d_{j'}$$

$$= \sum_{s=1}^{n} q_s d_s - q_j d_j + \sum_{j'>j} \frac{p_{v_j}}{1 - p_{v_j}} q_{j'} d_{j'}$$

$$= \tilde{f}_{\mathrm{os}}(p; i, h) - q_j d_j + \frac{p_{v_j}}{1 - p_{v_j}} \sum_{j'>j} q_{j'} d_{j'}.$$

When $p' = \mathrm{der}(v_j, 1; p)$, we have

$$\tilde{f}_{\mathrm{os}}(p'; i, h)$$

$$= p'_{v_1} d_1 + (1 - p'_{v_1})p'_{v_2} d_2 + \cdots + \left( \prod_{s=1}^{n-1}(1 - p'_{v_s}) \right) p'_{v_n} d_n$$

$$= \left( \sum_{s<j} \prod_{k=1}^{s-1}(1 - p_{v_k})p_{v_s} d_s \right) + \prod_{k'=1}^{j-1}(1 - p_{v'_k}) d_j$$

$$= \left( \sum_{s<j} \prod_{k=1}^{s-1}(1 - p_{v_k})p_{v_s} d_s \right) + \sum_{j' \ge j} \prod_{k'=1}^{j'-1}(1 - p_{v_{k'}})p_{v_{j'}} d_j$$

$$= \sum_{s<j} q_s d_s + \sum_{j' \ge j} q_{j'} d_j$$

$$= \sum_{s=1}^{n} q_s d_s + 0 + \sum_{j'>j} q_{j'}(d_j - d_{j'})$$

$$= \tilde{f}_{\mathrm{os}}(p; i, h) + \sum_{j'>j} q_{j'}(d_j - d_{j'}).$$

□

*Proof of Lemma 5.* The formula of the probability $\Pr_{X \sim p}[\{v_X \in V_X : (v_X, i) \in E\} = \emptyset]$ can be represented as the probability that none of $i$'s neighbor is chosen in $V_X$, which is $\prod_{v \in N_i}(1 - p_v)$. □

*Proof of Lemma 6.* If $(i, j) \notin E$, the value of $p_j$ does not affect $\tilde{f}_{\mathrm{cv}}(p; i)$ since $p_j$ is not involved in the value of $\tilde{f}_{\mathrm{cv}}(p; i)$. When $(i, j) \in E$, if $p' = \mathrm{der}(j, 0; p)$,

$$\prod_{v \in N_i}(1 - p'_v) = \prod_{v \in N_i, v \ne j}(1 - p_v) = \tilde{f}_{\mathrm{cv}}(p; i) - p_j \prod_{v \in N_i, v \ne j}(1 - p_v);$$

if $p' = \mathrm{der}(j, 1; p)$,

$$\prod_{v \in N_i} (1 - p'_v) = 0.$$

$\square$

*Proof of Lemma 7.* By linearity of expectation, we immediately have

$$\mathbb{E}_{X \sim p}[|\{(u_X, v_X) \in \binom{V_X}{2}\} \colon (u_X, v_X) \notin E|] = \sum_{(u,v) \in \binom{V}{2} \setminus E} \mathrm{Pr}_{X \sim p}[u, v \in V_X].$$

Regarding the inequality, it suffices to show that $|\{(u_X, v_X) \in \binom{V_X}{2}\} \colon (u_X, v_X) \notin E| \geq \mathbb{1}(V_X \text{ does not form a clique})$, which is true because $|\{(u_X, v_X) \in \binom{V_X}{2}\} \colon (u_X, v_X) \notin E| = 0$ if $V_X$ forms a clique, and $|\{(u_X, v_X) \in \binom{V_X}{2}\} \colon (u_X, v_X) \notin E| \geq 1$ otherwise. $\square$

*Proof of Lemma 8.* When $p' = \mathrm{der}(i, 0; p)$,

$$\tilde{f}_{cq}(p') = \sum_{(u,v) \in \binom{V}{2} \setminus E} p'_u p'_v = \sum_{(u,v) \in \binom{V}{2} \setminus E, u \neq i, v \neq i} p_u p_v = \tilde{f}_{cq}(p) - p_i \sum_{j \in [n], j \neq i, (i,j) \notin E} p_j.$$

When $p' = \mathrm{der}(i, 1; p)$,

$$\tilde{f}_{cq}(p') = \sum_{(u,v) \in \binom{V}{2} \setminus E, u \neq i, v \neq i} p_u p_v + \sum_{(i,j) \in \binom{V}{2} \setminus E} p_j = \tilde{f}_{cq}(p) + (1 - p_i) \sum_{j \in [n], j \neq i, (i,j) \notin E} p_j.$$

$\square$

*Proof of Theorem 3.* For any $p$ and $i$, we have

$$\tilde{g}(p)$$
$$= \mathbb{E}_{X \sim p} g(X)$$
$$= \sum_{X \in d^n} \mathrm{Pr}_p[X] g(X)$$
$$= \sum_{X \in d^n} \prod_{v \in [n]} p_{v X_v} g(X)$$
$$= \sum_{X \in d^n} \left( \prod_{v \in [n] \setminus \{i\}} p_{v X_v} \right) p_{i X_i} g(X)$$
$$= \sum_{r \in d} \sum_{X \colon X_i = r} \left( \prod_{v \in [n] \setminus \{i\}} p_{v X_v} \right) p_{i X_i} g(X)$$
$$= \sum_{r \in d} \sum_{X \colon X_i = r} \left( \prod_{v \in [n] \setminus \{i\}} p_{v X_v} \right) p_{ir} g(X)$$
$$= \sum_{r \in d} p_{ir} \sum_{X \colon X_i = r} \left( \prod_{v \in [n] \setminus \{i\}} p_{v X_v} \right) g(X)$$
$$= \sum_{r \in d} p_{ir} \sum_{X} \left( \prod_{v \in [n] \setminus \{i\}} p_{v X_v} \right) \mathbb{1}(X_i = r) g(X)$$
$$= \sum_{r \in d} p_{ir} \tilde{g}(\mathrm{der}(i, r; p))$$
$$\geq \sum_{r \in d} p_{ir} \tilde{g}(\mathrm{der}(i, r; p)),$$

completing the proof on entry-wise concavity. Regarding differentiability, since $\mathbb{E}_{X \sim p} g(X) = \sum_{X \in d^n} \mathrm{Pr}_p[X] g(X)$, it suffices to show that $\mathrm{Pr}_p[X] g(X) = \sum_{X \in d^n} \prod_{v \in [n]} p_{v X_v} g(X)$ is differentiable w.r.t $p$ for each $X \in \{0, 1\}^n$. Indeed, fix any $X$, $\sum_{X \in d^n} \prod_{v \in [n]} p_{v X_v} g(X)$ is a polynomial of $p_{ir}$'s, and is thus differentiable. $\square$

*Proof of Theorem 4.* See the proof for Theorem 2. It is easy to see that the reasoning still holds with $\{0, 1\}$ being replaced by any non-binary $d$. $\square$

*Proof of Lemma 9.* First we show $\tilde{g}_1(p) \geq \Pr_p[X \notin \mathcal{C}_1]$. Let $\mathcal{A}_1$ denote the event that at least one hard conflict is violated, and let $n_h$ denote the number of violated hard conflicts. By linearity of expectation, we have

$$\mathbb{E}_{X \sim p}[n_h] = \sum_{(u,v) \in E_h} \Pr_{X \sim p}[X_u = X_v].$$

For

$$\sum_{(u,v) \in E_h} \Pr_{X \sim p}[X_u = X_v] = \sum_{(u,v) \in E_h} \sum_{r=0}^{c-1} p_{ur} p_{vr},$$

it holds because $\Pr_{X \sim p}[X_u = X_v] = \sum_{r=0}^{c-1}[\text{both } u \text{ and } v \text{ have color } r] = p_{ur} p_{vr}$. Regarding the inequality, since $\Pr_{X \sim p}[\mathcal{A}_1] = \mathbb{E}_{X \sim p} \mathbb{1}(\mathcal{A}_1)$, if suffices to show that $n_h \geq \mathbb{1}(\mathcal{A}_1)$, which is true because $\mathcal{A}_1$ happens iff $n_h \geq 1$ by their definitions.

Then we show $\tilde{f}_{rc2}(p) \geq \mathbb{E}_{X \sim p} f_2(X)$. It is immediate since $\Pr_{X \sim p}[X_u = X_v] = \sum_{r=0}^{c-1}[\text{both } u \text{ and } v \text{ have color } r] = p_{ur} p_{vr}$. $\square$

*Proof of Lemma 10.* When $p' = \text{der}(i, x; p)$,

$$\begin{aligned}
\tilde{g}_1(\text{der}(i, x; p)) &= \sum_{(u,v) \in E_h} \sum_{r=0}^{c-1} p'_{ur} p'_{vr} \\
&= \sum_{(u,v) \in E_h} \sum_{r=0}^{c-1} p'_{ur} p'_{vr} \\
&= \sum_{(u,v) \in E_h, u \neq i, v \neq i} \sum_{r=0}^{c-1} p_{ur} p_{vr} + \sum_{(i,j) \in E_h} p_{jx} \\
&= \tilde{g}_1(p) + (1 - p_{ix}) \sum_{(i,j) \in E_h} p_{jx} - \sum_{x' \in d \setminus \{x\}} p_{ix'} \sum_{(i,j) \in E_h} p_{jx'} \\
&= \tilde{g}_1(p) + \sum_{x' \in d \setminus \{x\}} p_{ix'} \sum_{(i,j) \in E_h} (p_{jx} - p_{jx'}),
\end{aligned}$$

where $1 - p_{ix} = \sum_{x' \in d \setminus \{x\}} p_{ix'}$ has been used. Similarly,

$$\begin{aligned}
&\tilde{f}_2(\text{der}(i, x; p)) \\
&= -\sum_{e=(u,v) \in E_s} \sum_r p'_{ur} p'_{vr} \log(1 - P(e)) \\
&= -\sum_{e=(u,v) \in E_s, u \neq i, v \neq i} \sum_r p'_{ur} p'_{vr} \log(1 - P(e)) - \sum_{e=(i,j) \in E_s} p_{jx} \log(1 - P(e)) \\
&= \tilde{f}_2(p) - (1 - p_{ix}) \sum_{(i,j) \in E_s} p_{jx} \log(1 - P(i,j)) + \sum_{x' \in d \setminus \{x\}} p_{ix'} \sum_{(i,j) \in E_s} p_{jx'} \log(1 - P(i,j)) \\
&= \tilde{f}_2(p) + \sum_{x' \in d \setminus \{x\}} p_{ix'} \sum_{(i,j) \in E_s} (p_{jx'} - p_{jx}) \log(1 - P(i,j)).
\end{aligned}$$

$\square$

## D  ADDITIONAL TECHNICAL DETAILS

Here, we provide some additional technical details that are omitted in the main text.

## D.1 COMPUTATION OF THE POISSON BINOMIAL DISTRIBUTION

Here, we provide some implementation details on the computation of the Poisson binomial distribution, which is used in Section 4.1. We mainly follow the original paper (Hong, 2013) and an existing implementation online (Straka, 2017).

The main formula is $\Pr_{X \sim p}[|V_X| = t] = \frac{1}{n+1} \sum_{s=0}^{n} \exp(-\mathbf{i}\omega st) \prod_{j=1}^{n}(1 - p_j + p_j \exp(\mathbf{i}\omega s))$, where $\mathbf{i} = \sqrt{-1}$ and $\omega = \frac{2\pi}{n+1}$. See the original paper (Hong, 2013) for more technical details. For the incremental update, we use $(1 - p_i)^{-1} \sum_{s=0}^{t} q_{t-s} \left(\frac{p_i}{p_i - 1}\right)^s$ for $0 \leq p_i \leq 0.5$ and $(p_i)^{-1} \sum_{s=0}^{n-t-1} q_{t+s+1} \left(\frac{p_i - 1}{p_i}\right)^s$ for $0.5 < p_i \leq 1$, which results in higher numerical stability. See Lemma 2 and its Proof in Appendix C for more details.

# E ADDITIONAL THEORETICAL RESULTS

Here, we provide additional theoretical results.

## E.1 ADDITIONAL RESULTS ON NON-BINARY DECISIONS

Here, we provide additional theoretical results regarding non-binary decisions. Specifically, we extend the theoretical results in the existing works (Karalias & Loukas, 2020; Wang et al., 2022) to non-binary cases.

Recall the theoretical results (Theorem 5) by Karalias & Loukas (2020).

**Theorem 5** (Theorem 1 by Karalias & Loukas (2020)) Assume that $f$ is non-negative. Fix any $\beta > \max_{X \in \mathcal{C}} f(X; G)$, $\epsilon > 0$, and $t \in (0, 1]$ such that $(1 - t)\epsilon < \beta$. If $\tilde{f}(p_{\text{init}}; G) < \beta$, then $\Pr_{X \sim p_{\text{init}}}[f(X; G) < \epsilon \wedge X \in \mathcal{C}] \geq t$.

We extend Theorem 5 to non-binary cases.

**Theorem 7** (Non-binary extension of Theorem 5). Assume that $f$ is non-negative. Fix any $\beta > \max_{X \in \mathcal{C}} f(X; G)$, $\epsilon > 0$, and $t \in (0, 1]$ such that $(1 - t)\epsilon < \beta$. If $\tilde{f}(p_{\text{init}}; G) < \beta$, then $\Pr_{X \sim p_{\text{init}}}[f(X; G) < \epsilon \wedge X \in \mathcal{C}] \geq t$.

*Proof.* We shall follow the main idea in the original proof of Theorem 5 by Karalias & Loukas (2020), which is based on Markov's inequality. The key point is that the reasoning still holds when the decisions are non-binary. Specifically, we can define a probabilistic penalty function $\hat{f}(X; G) = f(X; G) + \beta \mathbb{1}(X \in \mathcal{C})$. Since $\beta > \max_{X \in \mathcal{C}} f(X; G)$, we have $\hat{f}(X; G) < \epsilon$ if and only if $f(X; G) < \epsilon$ and $X \in \mathcal{C}$. Therefore, using Markov's inequality, we have

$$\Pr_{X \sim p_{\text{init}}}[(f(X; G) < \epsilon) \wedge (X \in \mathcal{C})] = \Pr_{X \sim p_{\text{init}}}[\hat{f}(X; G) < \epsilon]$$
$$> 1 - \frac{1}{\epsilon} \mathbb{E}_{X \sim p_{\text{init}}}[\hat{f}(X; G)]$$
$$= 1 - \frac{1}{\epsilon} \mathbb{E}_{X \sim p_{\text{init}}}[f(X; G) + \beta \mathbb{1}(X \in \mathcal{C})]$$
$$> 1 - \frac{1}{\epsilon}(\beta)$$
$$> t.$$

$\square$

Recall the theoretical results (Theorem 6) by Wang et al. (2022).

**Theorem 6** (Theorem 1 by Wang et al. (2022)) If $\tilde{f}(p) \geq \mathbb{E}_{X \sim p} f(X) + \beta \Pr_{X \sim p}[X \notin \mathcal{C}], \forall p$ is entry-wise concave and non-negative with $\beta > \max(\tilde{f}(p_{\text{init}}), \max_{X \in \mathcal{C}} f(X))$, then for any permutation $\pi : [n] \rightarrow [n]$, starting from $p_{\text{cur}} = p_{\text{init}}$ and for $i \in [n]$ doing (1) $x^* \leftarrow \arg\min_{x \in \{0,1\}} \tilde{f}(\text{der}(\pi(i), x; p_{\text{cur}}))$ and (2) $p_{\text{cur}} \leftarrow \text{der}(i, x^*; p_{\text{cur}})$ will finally give a discrete $p_{\text{final}} \in \mathcal{C}$ such that $f(p_{\text{final}}) \leq \tilde{f}(p_{\text{init}})$.

We shall show that Theorem 6 can be extended to non-binary cases.

**Theorem 8** (Non-binary extension of Theorem 6). If $\tilde{f}(p) \geq \mathbb{E}_{X \sim p} f(X) + \beta \Pr_{X \sim p}[X \notin \mathcal{C}], \forall p$ is entry-wise concave and non-negative with $\beta > \max(\tilde{f}(p_{\text{init}}), \max_{X \in \mathcal{C}} f(X))$, then for any permutation $\pi : [n] \rightarrow [n]$, starting from $p_{\text{cur}} = p_{\text{init}}$ and for $i \in [n]$ doing (1) $x^* \leftarrow \arg\min_{x \in d=\{0,1,2,\ldots,c-1\}} \tilde{f}(\text{der}(\pi(i), x; p_{\text{cur}}))$ and (2) $p_{\text{cur}} \leftarrow \text{der}(i, x^*; p_{\text{cur}})$ will finally give a discrete $p_{\text{final}} \in \mathcal{C}$ such that $f(p_{\text{final}}) \leq \tilde{f}(p_{\text{init}})$.

*Proof.* We shall follow the main idea in the original proof of Theorem 6 by Wang et al. (2022), where the key idea was that entry-wise concavity ensures that local derandomization does not increase the objective. This key idea still holds with non-binary decisions. First, since after the series of local derandomization, for each $i$, it is locally derandomized exactly once, the final derandomized result should be discrete. Regarding $p_{\text{final}} \in \mathcal{C}$ and $f(p_{\text{final}}) \leq \tilde{f}(p_{\text{init}})$, we claim that "local derandomization does not increase the objective". Specifically, since $\tilde{f}$ is entry-wise concave, i.e., $\sum_{r \in d} p_{ir} \tilde{f}(\text{der}(i, r; p); G) \leq \tilde{f}(p; G), \forall G, p, i$, and $\sum_{r \in d} p_{ir} = 1$, we have $\min_{r \in d} \tilde{f}(\text{der}(i, r; p); G) \leq \sum_{r \in d} p_{ir} \tilde{f}(\text{der}(i, r; p); G) \leq \tilde{f}(p; G), \forall G, p, i$. Hence, indeed, "local derandomization does not increase the objective", and the final $f(X; G) + \beta \mathbb{1}(X \notin \mathcal{C}) \leq \tilde{f}(p_{\text{init}}) < \beta$, which implies that $f(X; G) \leq \tilde{f}(p_{\text{init}})$ and $\mathbb{1}(X \notin \mathcal{C}) = 0$, i.e., $X \in \mathcal{C}$, completing the proof. $\square$

# F ADDITIONAL PROBLEMS

The robust $k$-clique problem generalizes the maximum $k$-clique problem (Bomze et al., 1999) and it can be seen as an uncertain variant of the heaviest $k$-subgraph problem (Feige et al., 2001; Billionnet, 2005).

## F.1 ROBUST $k$-CLIQUE

**Definition.** Given (1) an uncertain graph $G = (V, E, P)$, and (2) $k \in \mathbb{N}$, we aim to find a subset of nodes $V_X \subseteq V$ such that (c1) $|V_X| = k$, (c2) $V_X$ forms a clique, and (c3) $\Pr[\text{all the edges between nodes in } V_X \text{ exist}]$ is maximized.

**Involved conditions:** (1) cardinality constraints, (2) cliques, and (3) uncertainty (see Sections 4.1, 4.4 & 4.6).

**Details.** Regarding conditions (c1)-(c2), we can directly use Lemmas 1 & 7. Regarding condition (c3), fix any $V_X$, the probability that all the edges between nodes in $V_X$ exist is $\prod_{(u,v) \in \binom{V_c}{2} \cap E} P_{uv}$. Maximizing the probability is equivalent to minimizing $f_1(X) := -\sum_{(u,v) \in \binom{V_c}{2} \cap E} \log P_{uv}$. We let $\hat{f}_1(X) := f_1(X)$ and let $\tilde{f}_1(p) := \mathbb{E}_{X \sim p} \hat{f}_1(X) = -\sum_{(u,v) \in E} p_u p_v \log P_{uv}$. The final objective is $\tilde{f}_{\text{RQ}}(p) = \tilde{f}_1(p) + \beta_1 \tilde{f}_{\text{cq}}(p) + \beta_2 \tilde{f}_{\text{card}}(p; \{k\})$ with constraint coefficients $\beta_1, \beta_2 > 0$.

Regarding the incremental differences, we only need to derive the incremental differences of $\tilde{f}_1$, which is $\Delta \tilde{f}_1(i, 1, p) = (p_i - 1) \sum_{v : (i,v) \in E} p_v \log P_{iv}$, and $\Delta \tilde{f}_1(i, 0, p) = -p_i \sum_{v : (i,v) \in E} p_v \log P_{iv}$.

## F.2 ROBUST DOMINATING SET

The robust dominating set problem generalizes the minimal dominating set problem Guha & Khuller (1998) and can also be seen as an uncertain version of set covering Caprara et al. (2000).

**Definition.** Given (1) an uncertain graph $G = (V, E, P)$, and (2) $k \in \mathbb{N}$, we aim to find a subset of nodes $V_X \subseteq V$ such that (c1) $|V_X| = k$, (c2) $V_X$ is a dominating set in the underlying deterministic graph, that is, for each $v \in V$, either $v \in V_X$ or $v$ has a neighbor in $V_X$, and (c3) the probability that $V_X$ is indeed a dominating set when considering the edge uncertainty, i.e. $\Pr[\bigwedge_{v \in V \setminus V_X} \bigvee_{u \in V_X} A_{uv}]$ is maximized. For each edge $(u, v) \in E$, $A_u v$ is the event that $(u, v)$ exists under edge certainty, which happens with probability $P_{uv}$.

**Involved conditions:** (1) cardinality constraints, (2) covered, and (3) uncertainty (see Sections 4.1, 4.3, & 4.6).

**Details.** Regarding conditions (c1), we can directly use Lemma 1. Specifically, $\tilde{f}_1(p) = \tilde{f}_{\text{card}}(p; \{k\})$.

Conditions (c2) and (c3) can be combined together. We first add self-loops on each node $v \in V$ (so that each node $v$ can cover $v$ itself), and then consider the condition as $X \in \mathcal{C}$ with $\mathcal{C} = \{X : \text{each node } v \in V \text{ is covered}\}$. Then we define $\hat{f}_2(X)$ as the expected number of nodes that are not covered (when taking the edge uncertain into consideration). It is easy to see that $\hat{f}_2(X) \geq \mathbb{1}(X \notin \mathcal{C}), \forall X \in \{0,1\}^n$. Note that here the uncertainty comes from the edge probabilities while the decisions are discrete. The formula of $\hat{f}_2$ is $\hat{f}_2(X) = \sum_{i \in V} \Pr[i \text{ is not covered}] = \sum_{i \in V \setminus V_X} \prod_{v \in N_i} (1 - P_{iv})$, where $N_i = \{v \in V : (i, v) \in E\}$ is the neighborhood of $i$. We then define $\tilde{f}_2(p) = \mathbb{E}_{X \sim p} \hat{f}_2(X)$, and its formula is $\tilde{f}_2(p) = \sum_{i \in V} \Pr[i \notin V_X] \prod_{v \in N_i} (1 - P_{iv}) = \sum_{i \in V} (1 - p_i) \prod_{v \in N_i} (1 - P_{iv})$. Combining all the conditions, the final probabilistic objective is $\tilde{f}_{\text{RDS}}(p) = \tilde{f}_2(p) + \beta \tilde{f}_1(p)$ with constraint coefficient $\beta > 0$.

Regarding the incremental differences, we only need to derive the incremental differences of $\tilde{f}_2$, which is $\Delta \tilde{f}_2(i, 1, p) = (p_i - 1) \prod_{v \in N_i} (1 - P_{iv})$ and $\Delta \tilde{f}_2(i, 0, p) = -p_i \prod_{v \in N_i} (1 - P_{iv})$.

## F.3 CLIQUE COVER

The clique cover problem (Gramm et al., 2009) is a classical NP-hard combinatorial problem. We consider its decision version, which is NP-complete.

**Definition.** Given (1) a graph $G = (V, E)$ and (2) $c \in \mathbb{N}$, we aim to partition the nodes into $c$ groups, such that each group forms a clique.

**Involved conditions:** (1) cliques and (2) non-binary decisions (see Sections 4.4 & 4.5).

**Details.** This is basically the non-binary extension of the "cliques" condition. For each $r \in d = \{0, 1, 2, \ldots, c - 1\}$, the condition holds for group-$r$ if the group is either empty or forms a clique. The group-$r$ is empty with probability $\prod_{i \in V} (1 - p_{ir})$, and we can use $\tilde{f}_{\text{cq}}(p_{\cdot, r}) \geq \Pr_{X \sim p}[\text{group-}r \text{ does not form a clique}]$, where $p_{\cdot, r} \in [0, 1]^n$ with $(p_{\cdot, r})_j = p_{j,r}$. Then the violation probability $\Pr[\text{violation}] = \Pr[\text{not empty} \wedge \text{does not form a clique}] \leq \Pr[\text{not empty}] + \Pr[\text{does not form a clique}]$. Therefore, we can have the final probabilistic objective $\tilde{f}_{\text{cc}}(p) = \sum_{r=0}^{c-1} 1 - \prod_{i \in V} (1 - p_{ir}) + \tilde{f}_{\text{cq}}(p_{\cdot, r})$. If we create a complete graph $K_V$ with self-loops on $V$, then $\prod_{i \in V} (1 - p_{ir})$ is $\tilde{f}_{\text{cv}}(p_{\cdot, r}; v, K_V)$ for any $v \in V$. Hence, we have $\tilde{f}_{\text{CC}}(p) = \sum_{r=0}^{c-1} 1 - \tilde{f}_{\text{cv}}(p_{\cdot, r}; v, K_V) + \tilde{f}_{\text{cq}}(p_{\cdot, r})$, and the incremental differences can be handled by those of $\tilde{f}_{cv}$ and $\tilde{f}_{cq}$.

## F.4 MINIMUM SPANNING TREE

The minimum spanning tree problem (Graham & Hell, 1985) is a classical combinatorial problem. Notably, it is not theoretically difficult and we have fast algorithms (Pettie & Ramachandran, 2002; Zhong et al., 2015) for the problem. But it is still interesting to see that our method can be applied to such a problem.

**Definition.** Given a graph $G = (V, E, W)$, we aim to find a subset of edges to form a connected tree (i.e., without cycles) containing all the nodes such that the total edge weights in the tree are minimized. Instead of considering choosing edges, we consider the decisions on nodes. Specifically, we put the nodes into different layers. Let $c \leq n$ be the number of layers, it is a non-binary problem, where each node $v$ is put into layer-$X_v$ with $X_v \in d = \{0, 1, 2, \ldots, c - 1\}$. For each node $v_\ell$ in layer $\ell > 0$, it would be connected to a parent $v_{prev}$ in the previous layer-$(\ell - 1)$ so that the edge weight of $(v_\ell, v_{prev})$ is minimized. The conditions are: (c1) each node is either in layer-0, or it can find a parent in the previous layer, and (c2) the total edge weights are minimized.

**Involved conditions:** (1) optimum w.r.t. a subset, (2) covered and (3) non-binary decisions (see Sections 4.2, 4.3, and 4.5).

**Details.** Regarding (c1), we let $\hat{f}_1$ be the number of nodes for which (c1) is violated. For each node $i$, it is in layer-0 with probability $p_{v0}$ and it can find at least one parent with probability $\sum_{\ell=1}^{c-1} \Pr[i \text{ is in layer-}\ell] \Pr[\text{at least one of } i\text{'s neighbors is in layer-}(\ell-1)] = \sum_{\ell=1}^{c-1} p_{i\ell}(1 - \prod_{v \in N_i}(1 - p_{v,\ell-1})) = \sum_{\ell=1}^{c-1} p_{i\ell}(1 - \tilde{f}_{\text{cv}}(p\cdot, \ell-1; i))$, where $p_{\cdot,\ell-1} \in [0,1]^n$ with $(p_{\cdot,\ell-1})_j = p_{j,\ell-1}$. Again, $N_i = \{v \in V \colon (i,v) \in E\}$ is the neighborhood of $i$. Note how the idea of "covered" is used here. Therefore, the probability that (c1) is violated for the node $i$ is $1 - p_{i0} - \sum_{\ell=1}^{c-1} p_{i\ell}(1 - \tilde{f}_{\text{cv}}(p\cdot, \ell-1; i))$. Now we are ready to compute $\tilde{f}_1(p) = \mathbb{E}_{X \sim p} \hat{f}_1(X) = \sum_{i \in V}(1 - p_{v0} - \sum_{\ell=1}^{c-1} p_{v\ell}(1 - \tilde{f}_{\text{cv}}(p\cdot, \ell-1; i)))$.

Regarding (c2), we use the idea of "optimum w.r.t. a subset". For a spanning tree, the total edge weights are $\sum_{i \in V \colon i \text{ not the root}} W(i, \text{the parent of } i)$. Note that in a minimum spanning tree, each non-root node should have a single parent. For each node $i$, the expected $W(i, \text{the parent of } i)$ is $\sum_{\ell=1}^{c-1} p_{i\ell} \tilde{f}_{\text{os}}(p_{;\ell-1}; i, W)$, where $p_{\cdot,\ell-1} \in [0,1]^n$ with $(p_{\cdot,\ell-1})_j = p_{j,\ell-1}$. The idea of "optimum w.r.t. a subset" has been used, where we consider the nodes being chosen into layer-$(\ell-1)$. Therefore, we have $\tilde{f}_2(p) = \sum_{i \in V} \sum_{\ell=1}^{c-1} p_{i\ell} \tilde{f}_{\text{os}}(p_{\cdot,\ell-1}; i, W)$. Combining the conditions, the final probabilistic objective is $\tilde{f}_{\text{MST}}(p) = \tilde{f}_2(p) + \beta \tilde{f}_1(p)$ with constraint coefficient $\beta > 0$. The incremental differences can be handled by those of $\tilde{f}_{\text{cv}}$ and $\tilde{f}_{\text{os}}$.

# G    COMPLETE EXPERIMENTAL SETTINGS AND RESULTS

Here, we provide detailed experimental settings and some additional experimental results.

## G.1    DETAILED EXPERIMENTAL SETTINGS

Here, we provide some details of the experimental settings.

### G.1.1    HARDWARE

All the experiments are run on a machine with two Intel Xeon® Silver 4210R (10 cores, 20 threads) processors, a 256GB RAM, and RTX2080Ti (11GB) GPUs. For the methods using GPUs, a single GPU is used.

### G.1.2    FACILITY LOCATION

Here, we provide more details about the settings of the experiments on the facility location problem. For the experiments on facility location and maximum coverage, we mainly follow the settings by Wang et al. (2023) and use their open-source implementation.[14]

**Datasets.** We consider both random synthetic graphs and real-world graphs:

- **Rand500:** We follow the way of generating random graphs by Wang et al. (2023). We generate 100 random graphs, where each graph contains 500 nodes. Each node $v$ has a two-dimensional location $(x_v, y_v)$, where $x_v$ and $y_v$ are sampled in $[0,1]$, independently, uniformly at random.
- **Rand800:** The rand800 graphs are generated in a similar way. The only difference is that each rand800 graph contains 800 nodes.
- **Starbucks:** The Starbucks datasets were used by Wang et al. (2023). We quote their descriptions as follows: "The datasets are built based on the project named Starbucks Location Worldwide 2021 version,[15] which is scraped from the open-accessible Starbucks store locator webpage.[16] We analyze and select 4 cities with more than 100 Starbucks stores, which are London (166 stores), New York City (260 stores), Shanghai (510 stores), and Seoul (569 stores). The locations considered are the real locations represented as latitude and longitude."

---

[14]https://github.com/Thinklab-SJTU/One-Shot-Cardinality-NN-Solver
[15]https://www.kaggle.com/datasets/kukuroo3/starbucks-locations-worldwide-2021-version
[16]https://www.starbucks.com/store-locator

- **MCD:** The MCD (McDonald's) dataset is available online.[17]. The dataset contains the locations of MCD branches in the United States. We divide the dataset into multiple sub-datasets by state, where each sub-dataset contains branches in the same state. We use the data from 8 states with the most ranches: CA (1248 branches), TX (1155 branches), FL (889 branches), NY (597 branches), PA (483 branches), IL (650 branches), OH (578 branches), and GA (442 branches).
- **Subway:** The Subway dataset is available online.[18] Similar to the MCD dataset, it contains the locations of subway branches in the United States. We also divide the dataset into multiple sub-datasets by state, where each sub-dataset contains branches in the same state. We use the data from 8 states with the most ranches: CA (2590 branches), TX (21994 branches), FL (1490 branches), NY (1066 branches), PA (865 branches), IL (1110 branches), OH (1171 branches), and GA (852 branches).
- For the real-world datasets, we use min-max normalization to make sure that each coordinate of each node (location) is also in $[0, 1]$ as in the random graphs.

**Inductive settings.** We follow the settings by Wang et al. (2023). For random graphs, the model is trained and tested on random graphs from the same distribution, but the training set and the test set are disjoint. For real-world graphs, the model is trained on the *rand500* graphs.

**Methods.** We consider both traditional methods and machine-learning methods:

- **Random:** Among all the locations, $k$ locations are picked uniformly at random; 240 seconds are given on each test graph.
- **Greedy:** deterministic greedy algorithms. We use the implementation of Wang et al. (2023).
- **Gurobi** (Gurobi Optimization, LLC, 2023) and **SCIP** (Bestuzheva et al., 2021; Perron & Furnon, 2023): The problems are formulated as MIPs and the two solvers are used; the time budget is set as 120 seconds, but the programs sometimes do not terminate until more time is used.
- **CardNN** (Wang et al., 2023): Three variants proposed in the original paper. We use the implementation of the original authors.
- **CardNN-noTTO**: In addition to training, CardNN also directly optimizes on each test graph in test time, and this is a variant of CardNN without test-time optimization. We use the implementation of the original authors.
- **EGN-naive**: EGN (Karalias & Loukas, 2020) with a naive objective construction and iterative rounding, which was used by Wang et al. (2023) as a baseline method. We use the derivation and implementation by Wang et al. (2022).
- **RL**: A reinforcement-learning method (Kool et al., 2019). We adapt the implementation by Berto et al. (2023).[19]

**Speed-quality trade-offs.** For the proposed method UCom2, we use test-time augmentation (Jin et al., 2023) on the test graphs by adding perturbations into both graph topology and features to obtain additional data. Specifically, we use edge dropout (Papp et al., 2021; Shu et al., 2022) and add Gaussian noise into features. The noise scale and the edge dropout ratios are both 0.2, which we do not fine-tune. The three variants of UCom2 are obtained by using different numbers of additional augmented data and taking the best objective. Specifically, the "short" version uses only the original test graphs, the "middle" version uses less time than CardNN-GS, and the "long" version uses less time than CardNN-HGS.

**Evaluation.** Given locations $(x_v, y_v)$'s for the nodes $v \in V$, if the final selected $k$ nodes are $v_1, v_2, \ldots, v_k$, the final objective is $\sum_{v \in V} \min_{i \in [k]} \mathrm{dist}(v_i, v)$, where the distance metric $\mathrm{dist}$ is the Euclidean squared distance used by Wang et al. (2023). We choose $k = 30$ locations in each graph, except for the rand800 graphs where we choose $k = 50$ locations.

**Hyperparameter fine-tuning.** For the proposed method UCom2 and the method CardNN by Wang et al. (2023), we conduct hyperparameter fine-tuning. For UCom2, we fine-tune the learning rate (LR) and constraint coefficient (CC). For CardNN, we fine-tune the training learning rate (LR)[20] and the Gumbel noise scale $\sigma$. For random graphs, we choose the best hyperparameter setting w.r.t. the

---

[17]https://www.kaggle.com/datasets/mdmdata/mcdonalds-locations-united-states

[18]https://www.kaggle.com/datasets/thedevastator/subway-the-fastest-growing-franchise-in-the-worl

[19]https://github.com/kaist-silab/rl4co

[20]CardNN uses (possibly) different learning rates for training and test-time optimization.

objective on the training set, because the distribution of the training set and the distribution of the test set are the same. For real-world graphs, we choose the smallest graph in each group of datasets as the validation graph, and we choose the best hyperparameter setting w.r.t. the objective on the validation graph.

We make sure that the number of candidate combinations (which is 15) is the same for both methods. Our hyperparameter search space is as follows:

- For UCoM2: LR $\in \{1e-1, 1e-2, 1e-3, 1e-4, 1e-5\}$ and CC $\in \{1e-1, 1e-2, 1e-3\}$
- For CardNN: LR $\in \{1e-1, 1e-2, 1e-3, 1e-4, 1e-5\}$ and $\sigma \in \{0.01, 0.15, 0.25\}$

Notably, after our fine-tuning, the performance of CardNN is at least the same and usually better than the performance using the hyperparameter settings in the open-source code of CardNN provided by the original authors. The best hyperparameter settings for each dataset are:

- Rand500:
    - UCoM2: LR $= 1e-2$, CC $= 1e-2$
    - CardNN: LR $= 1e-4$, $\sigma = 0.25$
- Rand800:
    - UCoM2: LR $= 1e-2$, CC $= 1e-2$
    - CardNN: LR $= 1e-4$, $\sigma = 0.25$
- Starbucks:
    - UCoM2: LR $= 1e-1$, CC $= 1e-2$
    - CardNN: LR $= 1e-4$, $\sigma = 0.15$
- MCD:
    - UCoM2: LR $= 1e-2$, CC $= 1e-2$
    - CardNN: LR $= 1e-5$, $\sigma = 0.25$
- Subway:
    - UCoM2: LR $= 1e-1$, CC $= 1e-2$
    - CardNN: LR $= 1e-5$, $\sigma = 0.01$

### G.1.3 MAXIMUM COVERAGE

Here, we provide more details about the settings of the experiments on the maximum coverage problem.

**Datasets.** We consider both random synthetic graphs and real-world graphs:

- **Rand500:** We follow the way of generating random graphs by Wang et al. (2023). Each item has a random weight chosen uniformly at random between 1 and 100. Each set contains a random number of items, and the number of items is chosen uniformly at random between 10 and 30. Each rand500 dataset contains 500 sets and 1000 items.
- **Rand1000:** The rand1000 graphs are generated in a similar way. The only difference is that each rand1000 dataset contains 1000 sets and 2000 items.
- **Twitch:** The Twitch datasets were used by Wang et al. (2023). We quote their descriptions as follows: "This social network dataset is collected by Rozemberczki et al. (2021) and the edges represent the mutual friendships between streamers. The streamers are categorized by their streaming language, resulting in 6 social networks for 6 languages. The social networks are DE (9498 nodes), ENGB (7126 nodes), ES (4648 nodes), FR (6549 nodes), PTBR (1912 nodes), and RU (4385 nodes). The objective is to cover more viewers, measured by the sum of the logarithmic number of viewers. We took the logarithm to enforce diversity because those top streamers usually have the dominant number of viewers."
- **Railway:** The railway datasets (Ceria et al., 1998) are available online.[21] The data were collected from real-world crew membership in Italian railways. We have three datasets: (1) rail507 with 507 sets and 63009 items, (2) rail516 with 516 sets and 47311 items, and (3) rail582 with 582 sets and 55515 items.

**Inductive settings.** We follow the settings by Wang et al. (2023). For random graphs, the model is trained and tested on random graphs from the same distribution, but the training set and the test set are disjoint. For real-world graphs, the model is trained on the *rand500* graphs.

---

[21]https://plato.asu.edu/ftp/lptestset/rail.

**Methods.** See the method descriptions above for the facility location problem in Appendix G.1.2.

**Speed-quality trade-offs.** See the descriptions above for the facility location problem in Appendix G.1.2.

**Evaluation.** Let $w_j$'s denote the weights of the items. The final objective is the summation of the weights of the covered items. An item $j$ is covered if at least one set containing $j$ is chosen. This is the term $\sum_{j \in T_X} W_j$ in Section 5.2.

**Hyperparameter fine-tuning.** The overall fine-tuning principles are the same as in the experiments on the facility location problem. See Appendix G.1.2.

Our hyperparameter search space is as follows:

- For UCom2: LR $\in \{1e-1, 1e-2, 1e-3, 1e-4, 1e-5\}$ and CC $\in \{10, 100, 500\}$
- For CardNN: LR $\in \{1e-1, 1e-2, 1e-3, 1e-4, 1e-5\}$ and $\sigma \in \{0.01, 0.15, 0.25\}$

The best hyperparameter settings for each dataset are:

- Rand500:
    - UCom2: LR $= 1e-5$, CC $= 500$
    - CardNN: LR $= 1e-5$, $\sigma = 0.15$
- Rand1000:
    - UCom2: LR $= 1e-5$, CC $= 500$
    - CardNN: LR $= 1e-5$, $\sigma = 0.15$
- Twitch:
    - UCom2: LR $= 1e-1$, CC $= 100$
    - CardNN: LR $= 1e-4$, $\sigma = 0.01$
- Railway:
    - UCom2: LR $= 1e-4$, CC $= 10$
    - CardNN: LR $= 1e-5$, $\sigma = 0.15$

### G.1.4 ROBUST COLORING

Here, we provide more details about the settings of the experiments on the robust coloring problem.

**Datasets.** We use four real-world uncertain graphs (Hu et al., 2017; Ceccarello et al., 2017; Chen et al., 2019). They are available online.[22] Some basic statistics of the datasets are as follows:

- **Collins:** $n = 1004$ nodes and $m = 8323$ edges; a deterministic greedy coloring algorithm uses 18 colors for the hard conflicts, and 36 colors for all the conflicts.
- **Gavin:** $n = 1727$ nodes and $m = 7534$ edges; a deterministic greedy coloring algorithm uses 7 colors for the hard conflicts, and 16 for all the conflicts.
- **Krogan:** $n = 2559$ nodes $m = 7031$ edges; a deterministic greedy coloring algorithm uses 8 colors for the hard conflicts, and 25 for all the conflicts.
- **PPI:** $n = 1912$ nodes $m = 22749$ edges; a deterministic greedy coloring algorithm uses 47 colors for the hard conflicts, and 53 for all the conflicts.

We take the largest connected component of each dataset. For each dataset, the $20\%$ edges with the highest edge weights are chosen as the hard conflicts.

**Methods.** We consider four baseline methods:

- **Greedy-RD:** The method first samples a random permutation of nodes, and then following the permutation, for each node, greedily chooses the best coloring to (1) avoid all the hard conflicts and (2) optimizes the objective; 300 seconds are given on each test graph.
- **Greedy-GA:** This is the method proposed by Yanez & Ramirez (2003) in the original paper of robust coloring. The difference between greedy-RD and greedy-GA is that greedy-GA uses a genetic algorithm (GA) to learn a good permutation instead of randomly sampling permutations; in the GA algorithm, the number of iterations is 20, the population size is 20, the crossover probability is 0.6, the mutation probability is 0.1, the elite ratio is 0.01, the parents proportion is 0.3.

---

[22]`https://github.com/Cecca/ugraph/tree/master/Reproducibility/Data`; `https://github.com/stasl0217/UKGE/tree/master/data`

- **Deterministic coloring (DC):** a deterministic greedy coloring algorithm (Kosowski & Manuszewski, 2004) is used to satisfy all the hard conflicts, and the soft conflicts are included in different random orders until no more soft conflicts can be satisfied. The maximum possible number of soft conflicts that can be included is found by binary search; 300 seconds are given on each test graph.
- **Gurobi**: the problem is formulated as an MIP and the solver is used; 300 seconds are given on each test graph.

**Hyperparameters.** For UCOM2, we do not fine-tune hyperparameters. We consistently use learning rate $\eta = 0.1$ and the constraint coefficient $\beta$ is set as the highest penalty on soft conflicts, i.e., $\max_{e=(u,v)\in E_s} \log(1 - P(e))$.

**Speed-quality trade-offs.** We record the running time of our method using only CPUs and using GPUs. For our method, we start from multiple random initial probabilities (each entry is sampled uniformly at random in $[0, 1]$), while making sure that even with only CPUs, our method uses less time than each baseline.

**Evaluation.** The recorded objective is the negative log-likelihood of no soft conflicts being violated, i.e., the function $f_2$ in Section 5.3.

## G.2 FULL RESULTS

Here, we provide the full raw results on each problem, together with the standard deviations of the results obtained by five random independent trials.

In Table 4, we provide the full raw results with standard deviations on the facility location problem.

In Table 5, we provide the full raw results with standard deviations on the maximum coverage problem.

## G.3 ABLATION STUDY

Here, we provide the results of the ablation study. First, we would like to point out that in the ex

### G.3.1 Q1: ARE PRINCIPLED PROBABILISTIC OBJECTIVES HELPFUL?

Here, we check whether the probabilistic objectives constructed following the principle proposed by us are helpful. We compare (a) EGN-naive (non-principled objectives and iterative rounding) and (b) UCOM2-iterative (principled objectives and iterative rounding). It is difficult to compare the full-fledged version of UCOM2 (principled objectives and greedy derandomization) and a variant with non-principled objectives and greedy derandomization, because we find computing the incremental differences of non-principled objectives nontrivial (yet less meaningful).

In Tables 6 and 7, we show the performance of EGN-naive and UCOM2-iterative on facility location and maximum coverage. We observe that in most cases, the optimization objective with the principled objectives is better. However, we also observe that using the principled objectives, the running time is sometimes higher. This is because the principled objective of cardinality constraints proposed by us is mathematically more complicated than the one used in EGN-naive formulated by Wang et al. (2023). This also validates the necessity of our fast incremental derandomization scheme, which can improve the speed.

### G.3.2 Q2: IS GREEDY DERANDOMIZATION BETTER THAN ITERATIVE ROUNDING?

Here, we check whether the proposed greedy derandomization is helpful, especially when compared to the iterative rounding proposed by Wang et al. (2022). We compare (a) the full-fledged version of UCOM2 (principled objectives and greedy derandomization) and (b) UCOM2-iterative (principled objectives and iterative rounding). In Tables 8 and 9, we show the performance of UCOM2 and UCOM2-iterative on facility location and maximum coverage.

We observe that when using (incremental) greedy derandomization (compared to iterative rounding), UCOM2 archives better optimization objectives within a shorter time, validating that the greedy derandomization scheme proposed by us is indeed helpful.

Table 4: Full raw results on facility location with the standard deviations. Running time (time): smaller the better. Objective (obj): smaller the better.

| method | rand500 | | rand800 | | starbucks | | mcd | | subway | |
|---|---|---|---|---|---|---|---|---|---|---|
| | obj↓ | time↓ | obj↓ | time↓ | obj↓ | time↓ | obj↓ | time↓ | obj↓ | time↓ |
| random | 3.41 | 240.00 | 3.48 | 240.00 | 0.54 | 240.00 | 1.60 | 240.00 | 2.80 | 240.00 |
| (std) | 0.008 | 0.000 | 0.004 | 0.000 | 0.004 | 0.000 | 0.011 | 0.000 | 0.007 | 0.000 |
| greedy | 2.85 | 2.10 | 2.67 | 5.88 | 0.35 | 6.51 | 1.12 | 11.51 | 1.99 | 26.00 |
| (std) | 0.000 | 0.012 | 0.000 | 0.025 | 0.000 | 0.032 | 0.000 | 0.054 | 0.000 | 0.115 |
| Gurobi | 2.60 | 121.65 | 2.80 | 124.11 | 0.31 | 102.82 | 1.30 | 124.62 | 4.61 | 131.35 |
| (std) | 0.118 | 0.206 | 0.052 | 0.181 | 0.005 | 0.944 | 0.073 | 0.766 | 1.032 | 0.193 |
| SCIP | 4.21 | 123.47 | 5.43 | 187.82 | 7.09 | 82.23 | 51.79 | 481.83 | 99.91 | 718.93 |
| (std) | 0.000 | 0.224 | 0.000 | 0.330 | 0.000 | 0.285 | 0.000 | 0.301 | 0.000 | 0.325 |
| CardNN-S | 2.75 | 13.06 | 2.45 | 13.17 | 0.52 | 15.20 | 1.87 | 2.01 | 1.94 | 21.59 |
| (std) | 0.070 | 0.446 | 0.089 | 0.385 | 0.060 | 0.165 | 0.095 | 0.299 | 0.187 | 1.840 |
| CardNN-GS | 2.41 | 74.01 | 2.34 | 142.94 | 0.43 | 69.81 | 1.09 | 39.97 | 2.02 | 31.39 |
| (std) | 0.024 | 1.395 | 0.066 | 1.612 | 0.046 | 1.483 | 0.075 | 1.304 | 0.145 | 0.380 |
| CardNN-HGS | 2.41 | 120.44 | 2.34 | 200.96 | 0.31 | 98.13 | 1.05 | 91.38 | 1.84 | 56.30 |
| (std) | 0.027 | 5.368 | 0.036 | 2.671 | 0.145 | 1.430 | 0.035 | 0.865 | 0.123 | 2.916 |
| CardNN-noTTO-S | 3.80 | 1.94 | 3.08 | 2.15 | 1.44 | 1.93 | 1.08 | 18.32 | 4.53 | 2.15 |
| (std) | 0.636 | 0.330 | 0.134 | 0.672 | 0.215 | 0.021 | 0.156 | 0.030 | 0.214 | 0.228 |
| CardNN-noTTO-GS | 2.75 | 28.77 | 2.65 | 52.48 | 0.42 | 32.24 | 1.06 | 62.64 | 2.04 | 7.82 |
| (std) | 0.039 | 0.355 | 0.294 | 0.325 | 0.194 | 0.239 | 0.145 | 0.261 | 0.226 | 0.121 |
| CardNN-noTTO-HGS | 2.74 | 42.95 | 2.34 | 200.96 | 0.46 | 27.21 | 1.19 | 28.14 | 1.99 | 45.74 |
| (std) | 0.031 | 1.225 | 0.093 | 3.262 | 0.215 | 0.423 | 0.102 | 2.989 | 0.148 | 0.039 |
| EGN-naive | 2.65 | 78.80 | 2.63 | 85.30 | 0.33 | 120.87 | 1.56 | 48.08 | 2.63 | 120.87 |
| (std) | 0.127 | 0.345 | 0.094 | 0.346 | 0.244 | 17.153 | 0.115 | 0.220 | 0.167 | 1.172 |
| RL-transductive | 5.57 | 300.00 | 5.18 | 300.00 | 2.97 | 1800.00 | 2.60 | 1800.00 | 4.50 | 1800.00 |
| (std) | 0.356 | 0.000 | 0.362 | 0.000 | 0.245 | 0.000 | 0.261 | 0.000 | 0.415 | 0.000 |
| RL-inductive | 4.07 | 300.06 | 4.27 | 300.54 | 0.79 | 300.04 | 2.40 | 300.04 | 4.23 | 300.05 |
| (std) | 0.227 | 0.019 | 0.143 | 0.157 | 0.148 | 0.014 | 0.241 | 0.016 | 0.395 | 0.015 |
| UСом2-short | 2.53 | 0.37 | 2.39 | 1.70 | 0.31 | 1.28 | 1.05 | 3.56 | 1.89 | 11.25 |
| (std) | 0.027 | 0.045 | 0.060 | 0.037 | 0.054 | 0.090 | 0.099 | 0.130 | 0.145 | 0.112 |
| UСом2-middle | 2.41 | 31.14 | 2.31 | 47.47 | 0.29 | 2.55 | 0.98 | 10.13 | 1.88 | 28.03 |
| (std) | 0.023 | 1.205 | 0.033 | 0.999 | 0.047 | 0.195 | 0.087 | 0.402 | 0.114 | 0.183 |
| UСом2-long | 2.41 | 62.28 | 2.31 | 81.51 | 0.29 | 10.12 | 0.95 | 42.33 | 1.79 | 50.42 |
| (std) | 0.023 | 1.702 | 0.034 | 1.087 | 0.046 | 0.540 | 0.087 | 0.858 | 0.114 | 0.334 |

In conclusion, each component in UCOM2 is helpful in most cases, but only when combining both principled objectives with greedy derandomization can we obtain the best synergy.

### G.3.3  Q3: DOES INCREMENTAL DERANDOMIZATION IMPROVE THE SPEED?

Here, we want to check how much the proposed incremental derandomization scheme using incremental differences helps in improving the speed. With greedy derandomization, we compare the running time of incremental derandomization and naive derandomization (i.e., evaluating the objective on each possible local derandomization case), on facility location and maximum coverage.

In Tables 10 and 11, we show the running time of UCOM2 when using incremental derandomization and when using naive derandomization, on facility location and maximum coverage.

We observe that using incremental derandomization significantly improves the derandomization speed, and the superiority is usually more significant when the dataset sizes increase.

### G.3.4  Q4: HOW DOES UCOM2 PERFORM WITH DIFFERENT CONSTRAINT COEFFICIENTS?

Here, we want to check how UCOM2 performs when using different constraint coefficients (i.e., different $\beta$ values) and fixing the other hyperparameters.

In Tables 12 to 14, we show the performance of UCOM2 when using different $\beta$ values, on facility location, maximum coverage, and robust coloring.

For facility location and maximum coverage, the candidate $\beta$ values are the same as in Appendices G.1.2 and G.1.3. We use the fastest version of UCOM2 without test-time augmentation.

Table 5: Full raw results on maximum coverage with the standard deviations. Running time (time): smaller the better. Objective (obj): larger the better.

| method | rand500 | | rand1000 | | twitch | | railway | |
|---|---|---|---|---|---|---|---|---|
| | obj↑ | time↓ | obj↑ | time↓ | obj↑ | time↓ | obj↑ | time↓ |
| random | 36638.16 | 240.00 | 70627.17 | 240.00 | 17383.20 | 240.00 | 7340.50 | 240.00 |
| (std) | 23.204 | 0.000 | 40.524 | 0.000 | 57.395 | 0.000 | 0.000 | 0.000 |
| greedy | 44312.81 | 0.09 | 88698.89 | 0.33 | 33822.40 | 0.69 | 7616.00 | 0.76 |
| (std) | 0.000 | 0.005 | 0.000 | 0.006 | 0.000 | 0.012 | 0.000 | 0.015 |
| Gurobi | 44932.06 | 120.07 | 88940.50 | 120.07 | 33840.40 | 0.61 | 7611.00 | 120.63 |
| (std) | 125.518 | 0.034 | 200.929 | 0.014 | 0.000 | 0.072 | 7.057 | 0.018 |
| SCIP | 43814.37 | 120.06 | 86282.30 | 119.50 | 33840.40 | 3.40 | 7600.00 | 121.13 |
| (std) | 498.180 | 0.018 | 215.140 | 0.129 | 0.000 | 0.281 | 0.000 | 0.073 |
| CardNN-S | 42012.77 | 11.60 | 83068.64 | 11.57 | 33834.60 | 8.11 | 7398.50 | 3.09 |
| (std) | 80.594 | 3.966 | 57.158 | 3.833 | 891.556 | 3.242 | 114.059 | 0.615 |
| CardNN-GS | 44736.29 | 38.85 | 88950.66 | 68.72 | 33840.40 | 16.57 | 7641.50 | 16.51 |
| (std) | 28.100 | 13.248 | 86.800 | 17.063 | 628.667 | 6.583 | 4.500 | 5.056 |
| CardNN-HGS | 44734.66 | 54.38 | 88980.52 | 88.52 | 33840.40 | 33.96 | 7633.00 | 31.46 |
| (std) | 32.200 | 0.880 | 86.800 | 0.163 | 0.000 | 0.210 | 0.000 | 0.494 |
| CardNN-noTTO-S | 33041.61 | 2.06 | 62906.71 | 1.67 | 109.60 | 1.07 | 7205.00 | 1.50 |
| (std) | 48.747 | 0.018 | 187.405 | 0.026 | 4.219 | 0.009 | 0.157 | 0.017 |
| CardNN-noTTO-GS | 37184.06 | 9.53 | 70623.51 | 19.19 | 1567.00 | 1.93 | 7332.00 | 5.93 |
| (std) | 95.720 | 0.798 | 70.779 | 0.052 | 697.403 | 0.024 | 4.530 | 0.052 |
| CardNN-noTTO-HGS | 37223.94 | 12.78 | 70685.54 | 24.60 | 1766.17 | 2.85 | 7351.67 | 6.30 |
| (std) | 59.084 | 1.223 | 165.656 | 0.180 | 602.371 | 0.231 | 5.977 | 0.817 |
| EGN-naive | 41378.43 | 120.72 | 81393.77 | 101.23 | 15448.20 | 120.72 | 7290.00 | 120.76 |
| (std) | 1054.832 | 2.098 | 264.491 | 0.173 | 36.515 | 0.709 | 50.569 | 1.548 |
| RL-transductive | 41461.65 | 300.00 | 73597.20 | 300.00 | 32143.20 | 1800.00 | 7307.00 | 1800.00 |
| (std) | 580.240 | 0.000 | 957.157 | 0.000 | 315.444 | 0.000 | 53.139 | 0.000 |
| RL-inductive | 34536.00 | 300.06 | 69155.00 | 300.18 | 19840.60 | 301.91 | 7320.50 | 301.78 |
| (std) | 22.154 | 0.017 | 42.216 | 0.022 | 55.146 | 0.254 | 9.578 | 0.195 |
| ours-short | 44608.67 | 0.91 | 89306.85 | 2.48 | 33828.40 | 1.82 | 7639.00 | 1.90 |
| (std) | 100.548 | 1.098 | 49.199 | 0.254 | 0.000 | 0.145 | 0.000 | 0.495 |
| ours-middle | 45002.78 | 14.69 | 89669.89 | 8.51 | 33828.40 | 11.43 | 7639.00 | 7.32 |
| (std) | 75.145 | 12.158 | 32.157 | 1.486 | 0.000 | 0.301 | 0.000 | 3.216 |
| ours-long | 45025.56 | 29.21 | 89896.96 | 78.15 | 33828.40 | 19.35 | 7644.50 | 14.22 |
| (std) | 74.968 | 19.156 | 32.116 | 10.159 | 0.000 | 0.578 | 0.601 | 5.745 |

Table 6: Ablation study on facility location: are principled probabilistic objectives helpful? Running time (time): smaller the better. Objective (obj): smaller the better.

| method | rand500 | | rand800 | | starbucks | | mcd | | subway | |
|---|---|---|---|---|---|---|---|---|---|---|
| | obj↓ | time↓ | obj↓ | time↓ | obj↓ | time↓ | obj↓ | time↓ | obj↓ | time↓ |
| EGN-naive | 2.65 | 78.80 | 2.63 | 85.30 | 0.33 | 120.87 | 1.56 | 48.08 | 2.63 | 120.87 |
| UCOM2-iterative | 2.65 | 169.56 | 2.67 | 205.12 | 0.33 | 162.15 | 1.05 | 87.45 | 1.96 | 52.37 |

Table 7: Ablation study on maximum coverage: are principled probabilistic objectives helpful? Running time (time): smaller the better. Objective (obj): larger the better.

| method | rand500 | | rand1000 | | twitch | | railway | |
|---|---|---|---|---|---|---|---|---|
| | obj↑ | time↓ | obj↑ | time↓ | obj↑ | time↓ | obj↑ | time↓ |
| EGN-naive | 41378.43 | 120.72 | 81393.77 | 101.23 | 15448.20 | 120.72 | 7290.00 | 120.76 |
| UCOM2-iterative | 42820.04 | 131.78 | 84397.79 | 209.95 | 16093.80 | 137.08 | 7304.50 | 120.21 |

For robust coloring, let the originally used $\beta_0 := \max_{e \in E_s} \log(1 - P(e))$, we consider three candidate values: $\frac{1}{2}\beta_0$, $\beta_0$, and $2\beta_0$. The other hyperparameters are fixed as the same.

Our observations are as follows. For facility location and maximum coverage:

- For random graphs, since the distribution of the training set and the distribution of the test set are the same, the originally used $\beta$ values perform well, usually the best among the candidates.

Table 8: Ablation study on facility location: is greedy derandomization better than iterative rounding? Running time (time): smaller the better. Objective (obj): smaller the better.

| method | rand500 | | rand800 | | starbucks | | mcd | | subway | |
|---|---|---|---|---|---|---|---|---|---|---|
| | obj↓ | time↓ | obj↓ | time↓ | obj↓ | time↓ | obj↓ | time↓ | obj↓ | time↓ |
| UCOM2-iterative | 2.65 | 169.56 | 2.67 | 205.12 | 0.33 | 162.15 | 1.05 | 87.45 | 1.96 | 52.37 |
| UCOM2-short | 2.53 | 0.37 | 2.39 | 1.70 | 0.31 | 1.28 | 1.05 | 3.56 | 1.89 | 11.25 |
| UCOM2-middle | 2.41 | 31.14 | 2.31 | 47.47 | 0.29 | 2.55 | 0.98 | 10.13 | 1.88 | 28.03 |
| UCOM2-long | 2.41 | 62.28 | 2.31 | 81.51 | 0.29 | 10.12 | 0.95 | 42.33 | 1.79 | 50.42 |

Table 9: Ablation study on maximum coverage: is greedy derandomization better than iterative rounding? Running time (time): smaller the better. Objective (obj): larger the better.

| method | rand500 | | rand1000 | | twitch | | railway | |
|---|---|---|---|---|---|---|---|---|
| | obj↑ | time↓ | obj↑ | time↓ | obj↑ | time↓ | obj↑ | time↓ |
| UCOM2-iterative | 42820.04 | 131.78 | 84397.79 | 209.95 | 16093.80 | 137.08 | 7304.50 | 120.21 |
| UCOM2-short | 44608.67 | 0.91 | 89306.85 | 2.48 | 33828.40 | 1.82 | 7639.00 | 1.90 |
| UCOM2-middle | 45002.78 | 14.69 | 89669.89 | 8.51 | 33828.40 | 11.43 | 7639.00 | 7.32 |
| UCOM2-long | 45025.56 | 29.21 | 89896.96 | 78.15 | 33828.40 | 19.35 | 7644.50 | 14.22 |

Table 10: Ablation study on facility location: does incremental derandomization improve the speed?

| | rand500 | rand800 | starbucks | mcd | subway |
|---|---|---|---|---|---|
| naive derandomization | 317.46 | 1061.02 | 231.85 | 1710.84 | 10196.05 |
| incremental derandomization | 0.37 | 1.70 | 1.28 | 3.56 | 11.25 |
| speed-up ratio | 849.65 | 623.30 | 180.77 | 480.14 | 906.30 |

Table 11: Ablation study on maximum coverage: does incremental derandomization improve the speed?

| | rand500 | rand1000 | twitch | railway |
|---|---|---|---|---|
| naive derandomization | 240.77 | 1186.06 | 2247.88 | 359.86 |
| incremental derandomization | 0.91 | 2.48 | 1.82 | 1.90 |
| speed-up ratio | 265.49 | 478.14 | 1231.81 | 189.52 |

Table 12: Ablation study on facility location: how does UCOM2 perform with different constraint coefficients? The results with the constraint coefficient originally used in our experiments are marked in bold. The numbers here are objectives (smaller the better).

| $\beta$ | rand500 | rand800 | starbucks | mcd | subway |
|---|---|---|---|---|---|
| 1e-1 | 2.50 | 2.47 | 0.31 | 1.02 | 1.75 |
| 1e-2 | **2.53** | **2.39** | **0.31** | **1.05** | **1.89** |
| 1e-3 | 3.19 | 2.79 | 1.85 | 1.41 | 3.83 |

- For real-world graphs, the originally used $\beta$ values do not achieve the best performance in some cases. In our understanding, this is because we use the smallest graph in each group of datasets as the validation graph, while the smallest graph possibly has a slightly different data distribution from the other graphs in the group, i.e., the test set.
- Overall, certain sensitivity w.r.t $\beta$ can be observed, but usually, multiple $\beta$ values can achieve reasonable performance.

For robust coloring:

- Overall, all the candidates $\beta$ vales can achieve similar performance.
- In other words, the performance of our method is not very sensitive to the value of $\beta$ on robust coloring.

Table 13: Ablation study on maximum coverage: how does UCOM2 perform with different constraint coefficients? The results with the constraint coefficient originally used in our experiments are marked in bold. The numbers here are objectives (larger the better).

| $\beta$ | rand500 | rand1000 | twitch | railway |
|---|---|---|---|---|
| 10 | 43744.80 | 87165.08 | 33801.80 | **7639.00** |
| 100 | 44382.36 | 88543.73 | **33828.40** | 7628.00 |
| 500 | **44608.67** | **89306.85** | 33825.80 | 7601.50 |

Table 14: Ablation study on robust coloring: how does UCOM2 perform with different constraint coefficients? The results with the constraint coefficient originally used in our experiments are marked in bold. The numbers here are objectives (smaller the better).

| $\beta$ | collins | | gavin | | krogan | | ppi | |
|---|---|---|---|---|---|---|---|---|
| | 18 colors | 25 colors | 8 colors | 15 colors | 8 colors | 15 colors | 47 colors | 50 colors |
| $\frac{1}{2}\beta_0$ | 78.32 | 15.61 | 46.56 | 6.70 | 52.04 | 0.87 | 2.93 | 1.01 |
| $\beta_0$ (originally used) | **82.26** | **15.16** | **42.99** | **6.72** | **52.44** | **0.87** | **2.93** | **1.01** |
| $2\beta_0$ | 81.17 | 15.83 | 44.96 | 6.77 | 55.25 | 0.87 | 2.93 | 1.01 |

# H DISCUSSIONS

## H.1 INDUCTIVE SETTINGS AND TRANSDUCTIVE SETTINGS

As discussed in Appendix A.2, the differentiable optimization in the pipeline can be done either in an inductive setting or in a transductive setting. Although ideally, a well-trained encoder can save much time without degrading the performance, in practice, inductive settings can be less effective (Li et al., 2023), especially when the training set and the test set have very different distributions (Drakulic et al., 2023).

As shown in our experimental results, the performance of CardNN (Wang et al., 2023) highly relies on test-time optimization (compare CardNN and CardNN-noTTO), which implies that the training is actually less essential than the direct optimization on test instances.

For UCOM2, we also observe that, when the training set and the test set are from different distributions, the training can be less helpful. Even applying derandomization on random probabilities can work well sometimes (but definitely not always).

## H.2 REINFORCEMENT LEARNING AND PROBABILISTIC-METHOD-BASED UL4CO

The connections between reinforcement learning and probabilistic-method-based UL4CO have been discussed by Wang et al. (2022). The direct connection comes from the fact that the policy gradient tries to approximate expectations by sampling, while probabilistic-method-based UL4CO aims to directly evaluate expectations. Differences also exist. In many cases, RL methods generate decisions in an autoregressive manner, while UL4CO methods try to do it in a one-shot manner (Wang et al., 2023), although one-shot RL has also been recently considered (Viquerat et al., 2023). Both the overhead of sampling and the autoregressive decision-encoding can potentially explain why UL4CO is usually more efficient than RL methods.

We focus on cases under complex conditions in this work. In RL, there are also similar subfields studying RL under constraints. On top of the basic difficulties of "sampling", constrained sampling for RL is even trickier and less efficient. Moreover, the analysis has been limited to simple constraints, e.g., linear and convex ones (Miryoosefi & Jin, 2022). We believe that this work shows that UL4CO is especially promising in cases under complex conditions.

