# OpenReview forum: "Unsupervised combinatorial optimization under complex conditions: Principled objectives and incremental greedy derandomization"
_ICLR.cc/2024/Conference — ICLR 2024 Conference Withdrawn Submission_

### Official Review · Reviewer_yrHB · 2023-10-31

**Soundness:** 3 good
**Presentation:** 2 fair
**Contribution:** 3 good
**Rating:** 6
**Confidence:** 3

**Summary:**

This paper considered the problem of unsupervised combinatorial optimization under complex conditions. The proposed UCOM2 consists of a principled probabilistic objective construction scheme and a derandomization scheme. The authors provided some theoretical results for the proposed scheme and applied them to various complex conditions, such as cardinality constraints and non-binary decisions. The authors also performed some experiments and showed that UCOM2 is general and practical.

**Strengths:**

Undoubtedly, this paper generalized some results of the references Karalias \& Loukas (2020) and Wang et al. (2022). To the reviewer's best understanding, the core contributions are two folds: (1) the author claimed the expectation $\tilde{f}:[0,1]^n\rightarrow \mathbb{R}$ of an optimization objective $f:\\{0,1\\}^n\rightarrow \mathbb{R}$, defined by $\mathbb{E}_{X\sim p}f(X)$, is differentiable and entry-wise concave with respect to $p$; (2) the authors conducted incremental greedy derandomization. Overall, this paper is well-written (except mathematical expressions are ugly organized, suggest using the align environment in Latex) and mathematically solid.

**Weaknesses:**

Unfortunately, some proofs of the core theorems are in questions.

(1) Page 16, Proof of Theorem 1: Please explain the fourth and fifth equality, i.e., first and second equalities as follows.
\begin{align}
&\sum\_X\prod\_{v\in V\_X}p\_v\prod\_{u\in[n]\setminus V\_X}(1-p\_u)g(X)\\\\
=&\sum\_X\prod\_{v\in V\_X,v\neq i}p\_v\prod\_{u\in[n]\setminus V\_X,v\neq i}(1-p\_u)(p\_ig({\rm{der}}(i,1;p))+(1-p\_i)g({\rm{der}}(i,0;p)))\\\\
=&p\_i\tilde{g}({\rm{der}}(i,1;p))+(1-p\_i)\tilde{g}({\rm{der}}(i,0;p))
\end{align}
The reviewer suspected that the first equality is wrong and the authors in fact showed $\tilde{g}$ is linear with respect to $p$.

(2) Page 17, Proof of Lemma 4: Please explain the following equalities,
\begin{align}
\tilde{f}\_{OS}(p^\prime;i,h)&=p^\prime\_{v\_1}d\_1+(1-p^\prime\_{v\_1})p^\prime\_{v\_2}d\_2+\cdots+(\prod\_{j=1}^{n-1}(1-p^\prime\_{v\_j}))p^\prime\_{v\_n}d\_n\\\\
&=\sum\_{j<i}\prod\_{k=1}^{j-1}(1-p\_{v\_k})p\_{v\_j}d\_j+0+\sum\_{j^\prime>i}\prod\_{1\leq k^\prime\leq j^\prime-1,k^\prime\neq i}(1-p\_{v\_k^\prime})p\_{v\_j^\prime}d\_{j^\prime}\\\\
&=\tilde{f}\_{OS}(p;i,h)-q\_jd\_j+\frac{p\_{v\_j}}{1-p\_{v\_j}}\sum\_{j^\prime>j}q\_{j^\prime}d\_{j^\prime}.
\end{align}
The reviewer suspected that the second equality is wrong.

(3) Page 17, Proof of Theorem 3: Please explain
\begin{align}
\sum\_{X\in d^n}(\prod\_{v\in [n]\setminus \\{i\\}}p\_{vX\_v})p\_{iX\_i}g(X)=\sum\_{r\in d}\tilde{f}({\rm{der}}(i,r;p))\leq \tilde{f}(p).
\end{align}

**Questions:**

Thanks the authors for providing the implemented codes. It would be great if the authors could provide readme.txt or demo codes that the reviewer can reproduce the experiment results. As for the experiments, the reviewer has one question, how to choose the parameter $\beta$? Have the authors done some ablation studies on the choices of $\beta$?

---

> ### Author Response · Authors · 2023-11-14
> **Response to Comment 1: Regarding Proof of Theorem 1**
>
> **Reviewer’s Comment:**
>
> - `Page 16, Proof of Theorem 1: Please explain the fourth and fifth equality, i.e., first and second equalities as follows.`
> \begin{align}
> &\sum_X\prod_{v\in V_X}p_v\prod_{u\in[n]\setminus V_X}(1-p_u)g(X)\\\\
> =&\sum_X\prod_{v\in V_X,v\neq i}p_v\prod_{u\in[n]\setminus V_X,v\neq i}(1-p_u)(p_ig({\rm{der}}(i,1;p))+(1-p_i)g({\rm{der}}(i,0;p)))\\\\
> =&p_i\tilde{g}({\rm{der}}(i,1;p))+(1-p_i)\tilde{g}({\rm{der}}(i,0;p))
> \end{align}
> - `The reviewer suspected that the first equality is wrong and the authors in fact showed` $\tilde{g}$ `is linear with respect to` $p$
>
>
> **Response:**
>
> - **Recall** **Theorem 1:** For any $g:\{0,1\}\to\mathbb R$, $\tilde{g}: [0,1]^n \to \mathbb R$ with $\tilde{g}(p)=\mathbb E_{X \sim p} g(X)$ is differentiable and entry-wise concave w.r.t $p$.
> - **Regarding the first equality:** Thanks for pointing out the unclear part. This part indeed did not have enough clarity with some notations ($g({\rm{der}}(i,0;p))$ and $g({\rm{der}}(i,1;p))$) not well defined. We have revised the equations and added more details:
>
> $$
> \begin{align}
> &\sum_{X} \prod_{v \in V_X} p_v \prod_{u \in [n] \setminus V_X} (1 - p_u) g(X) \\\\
>     =&\sum_{X \colon i \in V_X}\left(\prod_{v\in V_X,v\neq i}p_v\prod_{u\in[n]\setminus V_X}(1-p_u)\right) p_i g(X) + \sum_{X \colon i \notin V_X}\left(\prod_{v\in V_X}p_v\prod_{u\in[n]\setminus V_X,u\neq i}(1-p_u)\right) (1 - p_i) g(X) \\\\
>     =&p_i \sum_{X \colon i \in V_X}\prod_{v\in V_X,v\neq i}p_v\prod_{u\in[n]\setminus V_X}(1-p_u) g(X) + (1 - p_i) \sum_{X \colon i \notin V_X}\prod_{v\in V_X}p_v\prod_{u\in[n]\setminus V_X,u\neq i}(1-p_u) g(X)\\\\
>     =&p_i \tilde{g}(\operatorname{der}(i, 1; p)) + (1 - p_i) \tilde{g}(\operatorname{der}(i, 0; p))
> \end{align}
> $$
>
> - **Regarding $\tilde{g}$ is linear with respect to $p$:** The reviewer is correct and our statement is still valid. This is because linearity is a special case of entry-wise concavity. We only mentioned entry-wise concavity because that is what we need.

---

> > ### Comment · Reviewer_yrHB · 2023-11-14
> > **Response to the revised proof of Theorem 1**
> >
> > 1. Thanks for the revised proof. It's much clear now, but there are some small mistakes. The second $v\neq i$ above should be $u\neq i$.
> >
> > 2. The reviewer understand linearity is a special case of concavity. The question is why not saying "linear" rather than "concavity" in Theorem 1? Could all the analyses be true for entry-wise linear function $\tilde{f}$ throughout this paper?

---

> ### Author Response · Authors · 2023-11-14
> **Response to Comment 2: Regarding Proof of Lemma 4**
>
> **Reviewer’s Comment:**
>
> - `Page 17, Proof of Lemma 4: Please explain the following equalities,`
>  $$
> \begin{align}
> \tilde f_{\mathrm{os}}(p^\prime;i,h)&=p^\prime_{v_1}d_1+(1-p^\prime_{v_1})p^\prime_{v_2}d_2+\cdots+(\prod_{j=1}^{n-1}(1-p^\prime_{v_j}))p^\prime_{v_n}d_n \\\\
> &=\sum_{j<i}\prod_{k=1}^{j-1}(1-p_{v_k})p_{v_j}d_j+0+\sum_{j^\prime>i}\prod_{1\leq k^\prime\leq j^\prime-1,k^\prime\neq i}(1-p_{v_k^\prime})p_{v_j^\prime}d_{j^\prime}\\\\
> &=\tilde f_{\mathrm{os}}(p;i,h)-q_jd_j+\frac{p_{v_j}}{1-p_{v_j}}\sum_{j^\prime>j}q_{j^\prime}d_{j^\prime}.
> \end{align}
> $$
> - `The reviewer suspected that the second equality is wrong.`
>
> **Response:**
> - **Recall the current Lemma 4:** For any $p \in [0, 1]^n$ and $i \in [n]$, we define $q_i \coloneqq (\prod_{i' = 1}^{i-1} (1-p_{v_{i'}}))p_{v_{i}}$, the coefficient of $d_i$ in $\tilde f_{os}$. Then $\Delta \tilde f_{\mathrm{os}}(v_j, 0, p; i, h) = - q_j d_j + \frac{p_{v_j}}{1-p_{v_j}}\sum_{j' > j} q_{j'}d_{j'}$, and $\Delta \tilde f_{\mathrm{os}}(v_j, 1, p; i, h) = \sum_{j' > j} q_{j'}(d_j - d_{j'}), \forall j \in [n]$.
> - Thanks for pointing out the unclear part. **The final equation is correct**, while **some symbols** (e.g., $i$ and $j$) **were used twice with different meanings**, which possibly caused confusion. We have revised the notations and added more details:
> $$
> \begin{align}
> &\tilde f_{\mathrm{os}}(p^\prime; i,h) \\\\
> =& p^\prime_{v_1} d_1 + (1 - p^\prime_{v_1})p^\prime_{v_2} d_2 + \cdots + (\prod_{s = 1}^{n-1} (1-p^\prime_{v_s}))p^\prime_{v_{n}} d_n \\\\
> =& \sum_{s < j} \prod_{k = 1}^{s - 1} (1-p_{v_k})p_{v_{s}} d_s + 0 + \sum_{t > j} \prod_{1 \leq k \leq t - 1, k \neq j} (1-p_{v_{k}}) p_{v_{t}} d_{t}\\\\
> =& \sum_{s < j} q_s d_s + 0 + \sum_{j' > j} \frac{1}{1 - p_{v_j}} q_{j'} d_{j'} \\\\
> =& \sum_{s = 1}^{n} q_s d_s - q_jd_j + \sum_{j' > j} \frac{p_{v_{j}}}{1 - p_{v_{j}}} q_{j'} d_{j'} \\\\
> =& \tilde f_{\mathrm{os}}(p; i,h) - q_j d_j + \frac{p_{v_j}}{1-p_{v_j}}\sum_{j' > j} q_{j'}d_{j'}.
> \end{align}
> $$
>
> - We have also updated the notations in the statement of Lemma 4 accordingly: For any $p \in [0, 1]^n$ and $j \in [n]$, we define $q_j \coloneqq (\prod_{k = 1}^{j-1} (1-p_{v_{k}}))p_{v_{j}}$, the coefficient of $d_j$ in $\tilde f_{os}$. Then
> $\Delta \tilde f_{\mathrm{os}}(v_j, 0, p; i, h) = - q_j d_j + \frac{p_{v_j}}{1-p_{v_j}}\sum_{j^\prime > j} q_{j^\prime}d_{j^\prime}$, and $\Delta \tilde f_{\mathrm{os}}(v_j, 1, p; i, h) = \sum_{j^\prime > j} q_{j^\prime}(d_j - d_{j^\prime}), \forall j \in [n]$.

---

> > ### Comment · Reviewer_yrHB · 2023-11-18
> > **Response to the revised proof of Lemma 4**
> >
> > Thanks for your clarification. Unfortunately, there are some small mistakes. The third term in your second identity is wrong, i.e., $p_{v_k}d_{k}$ is wrong. Also, you haven't considered the case where $p_{v_j}=1$.

---

> ### Author Response · Authors · 2023-11-14
> **Response to Comment 3: Regarding Proof of Theorem 3**
>
> **Reviewer’s Comment:**
>
> - `Page 17, Proof of Theorem 3: Please explain`
>
>     $$
>     \begin{align}
>     \sum_{X\in d^n}(\prod_{v\in [n]\setminus \{i\}}p_{vX_v})p_{iX_i}g(X)=\sum_{r\in d}\tilde{f}({\rm{der}}(i,r;p))\leq \tilde{f}(p).
>     \end{align}
>     $$
>
>
> **Response:**
>
> - ********************Recall Theorem 3:******************** For any function $g: d^n \to \mathbb R$, $\tilde{g}: [0, 1]^{n \times c} \to \mathbb R$ with $\tilde{g}(p) = \mathbb E_{X \sim p} g(X)$ is differentiable and entry-wise concave, where $\mathbb E_{X \sim p} g(X) = \sum_{X \in d^n} \Pr_p[X] g(X)$ with $\Pr_p[X]= \prod_{v \in [n]} p_{vX_v}$.
> - Thanks for pointing out the unclear part. Our proof was indeed incomplete with some equations missing. We have revised our proof as follows:
> $$
> \begin{align}     &\tilde g(p) \\\\    =&\mathbb E_{X \sim p} g(X) \\\\    =&\sum_{X \in d^n} \Pr_{p}[X] g(X)\\\\    =&\sum_{X \in d^n} \prod_{v \in [n]} p_{vX_v} g(X)\\\\    =&\sum_{X \in d^n} (\prod_{v \in [n] \setminus \{i\}} p_{vX_v}) p_{iX_i} g(X)\\\\        =&\sum_{r \in d} \sum_{X \colon X_i = r} (\prod_{v \in [n] \setminus \{i\}} p_{vX_v}) p_{iX_i} g(X)\\\\    =&\sum_{r \in d} \sum_{X \colon X_i = r} (\prod_{v \in [n] \setminus \{i\}} p_{vX_v}) p_{ir} g(X)\\\\    =&\sum_{r \in d} p_{ir} \sum_{X \colon X_i = r}  (\prod_{v \in [n] \setminus \{i\}} p_{vX_v}) g(X)\\\\    =&\sum_{r \in d} p_{ir} \sum_{X}  (\prod_{v \in [n] \setminus \{i\}} p_{vX_v}) \mathbb{1}(X_i = r) g(X)\\\\    =&\sum_{r \in d} p_{ir} \tilde g(\operatorname{der}(i, r; p))       \end{align}
> $$

---

> > ### Comment · Reviewer_yrHB · 2023-11-18
> > **Response to the revised proof of Theorem 3**
> >
> > Thanks for the revised proof. The reviewer is wonder whether the condition, $\sum_{r\in d}p_{ir}\tilde{f}(der(i,r;p);G)\leq \tilde{f}(p;G)$, has been used. If not, how the authors claimed "completing the proof on P entry-wise concavity"?

---

> ### Author Response · Authors · 2023-11-14
> **Response to Comment 4: Regarding Using “Align” for Equations**
>
> **Reviewer’s Comment:**
>
> - `Mathematical expressions are ugly organized, suggest using the align environment in Latex`
>
> **Response:**
>
> - Thanks for the advice and the detailed review of our proofs. We have made equations in our proofs look better by using the “align” environment suggested by the reviewer.
> - In the main text, we currently do not have enough space due to the page limit, but we will try to reorganize the contents and beautify the equations for better readability in the camera-ready version.

---

> ### Author Response · Authors · 2023-11-14
> **Response to Comment 5: Regarding Code**
>
> **Reviewer’s Comment:**
>
> - `Thank the authors for providing the implemented codes. It would be great if the authors could provide readme.txt or demo codes so that the reviewer can reproduce the experiment results.`
>
> **Response:**
>
> - Thanks for pointing out the unclear part of our code.
> - In each folder, we already put a `training_cmd.txt` file explaining the code and providing example training commands, and the detailed hyperparameters can be found in Appendix G.1 (see the “hyperparameter fine-tuning” parts in G.1.2 and G.1.3).
> - We have further added a `readme.txt` file to provide overall guidance to readers. Please check the updated supplementary material.

---

> ### Author Response · Authors · 2023-11-14
> **Response to Comment 6: Regarding Constraint Coefficient $\beta$**
>
> **Reviewer’s Comment:**
>
> - `As for the experiments, the reviewer has one question, how to choose the parameter` $\beta$`? Have the authors done some ablation studies on the choices of` $\beta$`?`
>
> **Response:**
>
> - **********************************************************************************For facility location and maximum coverage:********************************************************************************** In Appendices G.1.2 and G.1.3, we discussed how we choose the hyperparameters for the proposed method UCom2 and the CardNN baseline method by Wang et al. (2023). For UCom 2, we fine-tune the learning rate and constraint coefficient (i.e., $\beta$).
>     - For random graphs, we choose the best hyperparameter setting w.r.t. the objective on the training set, because the distribution of the training set and the distribution of the test set are the same.
>     - For real-world graphs, we choose the smallest graph in each group of datasets as the validation graph, and we choose the best hyperparameter setting w.r.t. the objective on the validation graph. In each experiment, the training set, validation set, and test set are pairwise disjoint.
> - **********************************************************************************For robust coloring:********************************************************************************** We do not fine-tune the hyperparameters for the proposed method UCom2. Instead, we consistently use learning rate $\eta = 0.1$, and the constraint coefficient $\beta$ is set as the highest soft penalty, i.e., $\max_{e = (u, v) \in E_s} \log (1 - P(e))$.
> - We have further conducted ablation studies on the choices of $\beta$, we summarize our observations as follows:
>     - **********************************************************************************For facility location and maximum coverage:**********************************************************************************
>         - For random graphs, since the distribution of the training set and the distribution of the test set are the same, the originally used $\beta$ values perform well, usually the best among the candidates.
>         - For real-world graphs, the originally used $\beta$ values do not achieve the best performance in some cases. In our understanding, this is because we use the smallest graph in each group of datasets as the validation graph, while the smallest graph possibly has a slightly different data distribution from the other graphs in the group, i.e., the test set.
>         - Overall, certain **sensitivity** w.r.t $\beta$ can be observed, but usually, multiple $\beta$ values can achieve reasonable performance.
>     - **********************************************************************************For robust coloring:**********************************************************************************
>         - Overall, all the candidate $\beta$ values can achieve similar performance.
>         - In other words, the performance of our method is not very sensitive to the value of $\beta$ on robust coloring.
> - The detailed results are as follows. The numbers here are objectives, and **the results with the originally used constraint coefficient (NOT the best results) are marked in bold**.
>
> | facility location (smaller the better) | rand500 | rand800 | starbucks | mcd | subway |
> | --- | --- | --- | --- | --- | --- |
> | $\beta = 1e-1$ | 2.50 | 2.47 | 0.31 | 1.02 | 1.75 |
> | $\beta = 1e-2$ | **2.53** | **2.39** | **0.31** | **1.05** | **1.89** |
> | $\beta = 1e-3$ | 3.19 | 2.79 | 1.85 | 1.41 | 3.83 |
>
> | maximum coverage (larger the better) | rand500 | rand1000 | twitch | railway |
> | --- | --- | --- | --- | --- |
> | $\beta = 10$ | 43744.80 | 87165.08 | 33801.80 | **7639.00** |
> | $\beta = 100$ | 44382.36 | 88543.73 | **33828.40** | 7628.00 |
> | $\beta = 500$ | **44608.67** | **89306.85** | 33825.80 | 7601.50 |
>
> | robust coloring (smaller the better) | collins | collins | gavin | gavin | krogan | krogan | ppi | ppi |
> | --- | --- | --- | --- | --- | --- | --- | --- | --- |
> | constraint coefficient $\beta$ | 18 colors | 25 colors | 8 colors | 15 colors | 8 colors | 15 colors | 47 colors | 50 colors |
> | $\frac{1}{2}\beta_0$ | 78.32 | 15.61 | 46.56 | 6.70 | 52.04 | 0.87 | 2.93 | 1.01 |
> | $\beta_0$ (originally used) | **82.26** | **15.16** | **42.99** | **6.72** | **52.44** | **0.87** | **2.93** | **1.01** |
> | $2\beta_0$ | 81.17 | 15.83 | 44.96 | 6.77 | 55.25 | 0.87 | 2.93 | 1.01 |
>
> - We have included the ablation studies in the revised manuscript too. See Appendix G.3.4 in the revised manuscript for more details.

---

> > ### Comment · Reviewer_yrHB · 2023-11-20
> >
> > Thanks for conducting the ablation studies for the hyperparameter $\beta$. (This could be a naive question.) Why not the authors choose a larger graph as the validation graph to increase the stability of the performance for real-world graphs? How small is the graph (number of vertices and edges) that the authors choose compared with other training graphs?

---

> > > ### Author Response · Authors · 2023-11-20
> > > **RE: RE: Response to Comment 6: Regarding Constraint Coefficient $\beta$ (Part 1/2)**
> > >
> > > **Question:** `Why not the authors choose a larger graph as the validation graph to increase the stability of the performance for real-world graphs?`
> > >
> > > **Response:**
> > >
> > > - It is actually a good question, which enriches our empirical evaluations.
> > > - In our previous experiments, we simply set a deterministic standard for choosing validation graphs to ensure a fair comparison and reduce randomness. The validation graph can be the smallest graph as we did, and it can also be a larger graph, as the reviewer suggested.
> > > - We thank the reviewer for the suggestion. We have tried using the largest graph in each group as the validation graph too. We observed that the best hyperparameters are different in many cases, and the performance of our method increases overall (see the detailed results below), showing that the performance of our method is robust to different choices of validation graphs.
> > >
> > > | problem: facility location | left: learning rate | right: constraint coefficient $\beta$ |
> > > | --- | --- | --- |
> > > | dataset group | using the smallest graph | using the largest graph |
> > > | starbucks | (1e-1, 1e-2) | (1e-2, 1e-1) |
> > > | mcd | (1e-2, 1e-2) | (1e-4, 1e-1) |
> > > | subway | (1e-1, 1e-2) | (1e-4, 1e-1) |
> > >
> > > | problem: maximum coverage | left: learning rate | right: constraint coefficient $\beta$ |
> > > | --- | --- | --- |
> > > | dataset group | using the smallest graph | using the largest graph |
> > > | twitch | (1e-1, 100) | (1e-1, 100) |
> > > | railway | (1e-4, 10) | (1e-2, 10) |
> > > - In most cases (expect for the “railway” datasets), we found the reviewer’s intuition correct. Indeed, we observed that using the largest graph as the validation graph usually improves the performance of our method for real-world graphs, compared to when we used the smallest graph for validation.
> > >     - Therefore, when using the largest graph as the validation graph, the overall superiority of our method remains, or even increases.
> > > - Below are the detailed results of our method when the validation graph is the smallest graph (our previous results) and the largest graph (additionally added results).
> > >     - For a fair comparison, in each dataset group, both the smallest and the largest graphs are excluded when we calculate the performance below.
> > >     - Note that in our current manuscript, the results are calculated only excluding the smallest graph in each group (i.e., the validation graph we used), so the results below look different from the ones we currently have in the manuscript.
> > >     - The results on “twitch” are omitted because the hyperparameters validated on the smallest graph and the largest graph are the same.
> > >
> > > | dataset group: starbucks | obj (smaller = better) | time (smaller = better) |
> > > | --- | --- | --- |
> > > | UCom2-short (smallest) | 0.255 | 1.147 |
> > > | UCom2-middle (smallest) | 0.238 | 2.199 |
> > > | UCom2-long (smallest) | 0.237 | 8.343 |
> > > | UCom2-short (largest) | 0.236 | 1.000 |
> > > | UCom2-middle (largest) | 0.239 | 1.744 |
> > > | UCom2-long (largest) | 0.228 | 8.336 |
> > >
> > > | dataset group: mcd | obj (smaller = better) | time (smaller = better) |
> > > | --- | --- | --- |
> > > | UCom2-short (smallest) | 1.078 | 3.054 |
> > > | UCom2-middle (smallest) | 1.002 | 7.766 |
> > > | UCom2-long (smallest) | 0.975 | 32.453 |
> > > | UCom2-short (largest) | 0.993 | 1.969 |
> > > | UCom2-middle (largest) | 0.976 | 6.447 |
> > > | UCom2-long (largest) | 0.966 | 27.883 |
> > >
> > > | dataset group: subway | obj (smaller = better) | time (smaller = better) |
> > > | --- | --- | --- |
> > > | UCom2-short (smallest) | 1.841 | 7.615 |
> > > | UCom2-middle (smallest) | 1.773 | 16.968 |
> > > | UCom2-long (smallest) | 1.716 | 33.132 |
> > > | UCom2-short (largest) | 1.732 | 7.565 |
> > > | UCom2-middle (largest) | 1.695 | 19.553 |
> > > | UCom2-long (largest) | 1.681 | 33.774 |
> > >
> > > | dataset group: railway | obj (larger = better) | time (smaller = better) |
> > > | --- | --- | --- |
> > > | UCom2-short (smallest) | 7713.00 | 1.820 |
> > > | UCom2-middle (smallest) | 7713.00 | 6.831 |
> > > | UCom2-long (smallest) | 7724.00 | 13.161 |
> > > | UCom2-short (largest) | 7658.00 | 1.822 |
> > > | UCom2-middle (largest) | 7663.00 | 7.019 |
> > > | UCom2-long (largest) | 7682.00 | 13.161 |

---

> > > > ### Author Response · Authors · 2023-11-20
> > > > **RE: RE: Response to Comment 6: Regarding Constraint Coefficient $\beta$ (Part 2/2)**
> > > >
> > > > **Question:** `How small is the graph (number of vertices and edges) that the authors choose compared with other training graphs?`
> > > >
> > > > **Response:**
> > > >
> > > > - We verbally described the graphs used in our experiments in our manuscript. See the “datasets” parts in Appendices G.1.2 and G.1.3.
> > > > - The smallest graph in each dataset group is not always absolutely small. In such cases, even the smallest graph is larger than the training graphs.
> > > > - Let us further summarize the basic statistics (e.g., the number of vertices and the number of edges) of the random graphs (training graphs) and the real-world graphs (validation/test graphs) used in our experiments on real-world graphs.
> > > >     - In each group, the datasets are sorted w.r.t. the number of nodes (from small to large).
> > > >     - In our previous experiments, in each group, we used the smallest graph, i.e., the first one below the training set in each table.
> > > >     - In our additional experiments following the suggestion of the reviewer, in each group, we used the largest graph, i.e., the last one in each table.
> > > >
> > > > | dataset group: starbucks | problem: facility location |  |
> > > > | --- | --- | --- |
> > > > | dataset | # vertices = # locations | # edges |
> > > > | rand500 (training set) | 500 | 18153.82±582.55 |
> > > > | london | 166 | 12502 |
> > > > | newyork | 260 | 33982 |
> > > > | shanghai | 510 | 112156 |
> > > > | seoul | 569 | 160193 |
> > > >
> > > > | dataset group: subway | problem: facility location |  |
> > > > | --- | --- | --- |
> > > > | dataset | # vertices = # locations | # edges |
> > > > | rand500 (training set) | 500 | 18153.82±582.55 |
> > > > | GA | 852 | 192240 |
> > > > | PA | 865 | 86473 |
> > > > | NY | 1066 | 296316 |
> > > > | IL | 1110 | 324474 |
> > > > | OH | 1171 | 293473 |
> > > > | FL | 1490 | 445762 |
> > > > | TX | 2194 | 879290 |
> > > > | CA | 2590 | 2328764 |
> > > >
> > > > | dataset group: mcd | problem: facility location |  |
> > > > | --- | --- | --- |
> > > > | dataset | # vertices = # locations | # edges |
> > > > | rand500 (training set) | 500 | 18153.82±582.55 |
> > > > | GA | 442 | 50486 |
> > > > | PA | 483 | 30843 |
> > > > | OH | 578 | 41524 |
> > > > | NY | 597 | 90071 |
> > > > | IL | 650 | 126748 |
> > > > | FL | 889 | 159877 |
> > > > | TX | 1155 | 256689 |
> > > > | CA | 1248 | 545932 |
> > > >
> > > > | dataset group: twtich | problem: maximum coverage |  |  |  |
> > > > | --- | --- | --- | --- | --- |
> > > > | dataset | # sets | # items | # vertices = # sets + # items | # edges |
> > > > | rand500 (training set) | 500 | 1000 | 1500 | 9991.7±140.34 |
> > > > | PTBR | 1912 | 1912 | 3824 | 31299 |
> > > > | RU | 4385 | 4385 | 8770 | 37304 |
> > > > | ES | 4648 | 4648 | 9296 | 59382 |
> > > > | FR | 6549 | 6549 | 13098 | 112666 |
> > > > | ENGB | 7126 | 7126 | 14252 | 35324 |
> > > > | DE | 9498 | 9498 | 18996 | 153138 |
> > > >
> > > > | dataset group: rail | problem: maximum coverage |  |  |  |
> > > > | --- | --- | --- | --- | --- |
> > > > | dataset | # sets | # items | # vertices = # sets + # items | # edges |
> > > > | rand500 (training set) | 500 | 1000 | 1500 | 9991.7±140.34 |
> > > > | rail516.txt | 400 | 47311 | 47711 | 80163 |
> > > > | rail507.txt | 400 | 63009 | 63409 | 80163 |
> > > > | rail582.txt | 400 | 55515 | 55915 | 80163 |

---

> > > > > ### Author Response · Authors · 2023-11-22
> > > > >
> > > > > Dear Reviewer yrHB,
> > > > >
> > > > > Thank you for your detailed review again! Our work has improved regarding both theoretical analysis and empirical evaluation because of your constructive comments.
> > > > >
> > > > > We are wondering after our fruitful discussions, how much our responses have addressed your concerns. If you have any further questions you would like us to elaborate on, please let us know. We value your insights and are open to further discussion to enhance our work.
> > > > >
> > > > > Thank you once again for your time.
> > > > >
> > > > > Best,
> > > > >
> > > > > Submission1650 Authors

---

> > > > > > ### Comment · Reviewer_yrHB · 2023-11-22
> > > > > >
> > > > > > Thanks for your hard work. The reviewer is very appreciate for the authors' work and discussions. The rating has been changed to "marginally above the acceptance threshold". All the best!

---

> > > > > > > ### Author Response · Authors · 2023-11-22
> > > > > > >
> > > > > > > Dear Reviewer yrHB,
> > > > > > >
> > > > > > > Thank you for acknowledging the improvement of our work via our discussions.
> > > > > > > We once again thank you for your time.
> > > > > > >
> > > > > > > Best,
> > > > > > >
> > > > > > > Submission1650 Authors

---

> ### Author Response · Authors · 2023-11-14
> **Response to Comment 7: Regarding Contributions**
>
> **Reviewer’s Comment:**
>
> - `The core contributions are two folds: (1) the author claimed the expectation` $\tilde{f}:[0,1]^n\rightarrow \mathbb{R}$ `of an optimization objective` $f:\{0,1\}^n\rightarrow \mathbb{R}$`, defined by` $\mathbb{E}_{X\sim p}f(X)$`, is differentiable and entry-wise concave with respect to` $p$`; (2) the authors conducted incremental greedy derandomization.`
>
> **Response:**
>
> - Thanks for your detailed review and your acknowledgment of our contributions, especially theoretical contributions. We would like to further clarify our contributions.
> - Our contributions regarding objective construction are **more than just theoretical results** (i.e., Theorems 1 and 3), but also include a practical and principle scheme, as well as the derivations for the considered complex conditions. In other words, we not only prove that such good probabilistic objectives exist, but also (1) show how to construct them in general and (2) showcase derivations for specific cases. Let us elaborate below.
> - **Our derivations on the considered complex conditions are nontrivial.** In Section 4, for each considered complex condition, we derive a **probabilistic objective** and **incremental updates** that abide by the proposed schemes. Such derivations (see Lemmas 1-8 and their proofs) are nontrivial, containing solid technical contributions. Specifically, the results by Karalias & Loukas (2020) and Wang et al. (2022) cannot produce such derivations.
> - **Our schemes provide practical general guidelines.** Using the results by Karalias & Loukas (2020) and Wang et al. (2022), when encountering a combinatorial optimization (CO) problem with complex conditions (e.g., any problem considered in this work), one can only know what properties a good probabilistic objective should satisfy, but do not have any practice guideline for how to construct a good one. By contrast, our scheme provides a practical guideline for any problem:
>     - If the encountered problem involves any complex condition considered in this work, our derivations can be immediately used. We have shown the generality of the considered conditions by various problems in Section 5 and Appendix F. That is, we expect that the considered conditions are involved in many problems.
>     - Even if the encountered problem involves some conditions that are not considered in this work, one can still follow our template in Section 4 to construct probabilistic objectives (and incremental updates) for those conditions.
>     - After dealing with each condition involved in the encountered problem, one can always follow our template in Section 5 to obtain a theoretically desirable probabilistic objective (and incremental updates) for the whole problem.

---

> ### Author Response · Authors · 2023-11-14
> **RE: Response to the revised proof of Theorem 1**
>
> Dear Reviewer yrHB,
>
> 1. Thanks for your reply! We have just corrected that typo.
> 2. We only mentioned entry-wise concavity because that is what we need for Target 1, especially the good property (P4). We can definitely state the "linearity" in Theorem 1, but this does not enable us to have stronger claims for our other theoretical results. Importantly, yes, all the analyses throughout this paper are still true for entry-wise linear functions. More specifically, all the analyses only require concavity, having a stronger condition (i.e., linearity) is definitely fine.
>
> If there are still unclear points, please let us know!
>
> Best,
>
> Submission1650 Authors

---

> ### Author Response · Authors · 2023-11-18
> **RE: Response to the revised proof of Lemma 4**
>
> Dear Reviewer yrHB,
>
> Really thank you for your careful review. We are sorry for being careless and for the small typos.
> 1. You are right! The $p_{v_k} d_k$ should be $p_{v_t} d_t$. We have corrected that in the updated manuscript.
> 2. When $p_{v_j} = 1$, we would indeed have "division by zero" problems.
> - We have considered that and we have covered that special case in footnote 6 in the main text (see page 5).
> - In practice (i.e., in our implementation), we use a small $\epsilon > 0$ and make sure that each $p_i \in [\epsilon, 1 - \epsilon]$ for numerical stability. We have further clarified this in the updated manuscript (see the "implementation details" in Section 4.1 and the updated footnote 6).
>
> We sincerely thank you for your detailed review again, which really improves the quality of the mathematical language in our manuscript.
>
> Best,
>
> Submission1650 Authors

---

> ### Author Response · Authors · 2023-11-18
> **RE: Response to the revised proof of Theorem 3**
>
> Dear Reviewer yrHB,
>
> Thanks for your reply! Let us clarify it. As mentioned in Section 4.5:
> - For non-binary cases, a function $\tilde{f}: [0, 1]^{n \times c} \to \mathbb R$ is *entry-wise concave* if $\sum_{r \in d} p_{ir} \tilde{f}(\operatorname{der}(i, r; p); G) \leq \tilde{f}(p; G), \forall G, p, i$.
> - In Theorem 3, we want to prove that this function $\tilde g$ is entry-wise concave. Note that the symbols are different in the definition ($\tilde f$) and in the statement ($\tilde g$).
> - Therefore, we need to prove that $\sum_{r \in d} p_{ir} \tilde{g}(\operatorname{der}(i, r; p); G) \leq \tilde{g}(p; G), \forall G, p, i$.
> - In our proof, we show that for each $p$ and $i$, $\tilde{g}(p) = \sum_{r \in d} p_{ir} \tilde{g}(\operatorname{der}(i, r; p))$, which, similar to the situation in Theorem 1, implies that $\tilde{g}(p) \geq \sum_{r \in d} p_{ir} \tilde{g}(\operatorname{der}(i, r; p))$, completing the proof for entry-wise concavity. We have further clarified this in our updated manuscript.
>
> In our understanding, our previous statement and proof might be slightly confusing in that
> 1. We did not mention $G$ in our proof, but we had $G$ in our definition of entry-wise concavity.
> - We have removed $G$ in our definition of entry-wise concavity, since we realized that the definition of a function $[0, 1]^{n \times c} \to \mathbb R$ (or $[0, 1]^{n} \to \mathbb R$ for binary cases) should not depend on a specific input graph $G$.
> - We have revised the related definitions in our updated manuscript (see Sections 2.2.2 and 4.5).
> 2. Inequality ($\leq$) was used in the statement, while we showed equality ($=$) in our proof.
> - This is a similar issue as we have discussed for Theorem 1. Equality is a special case of, and thus stronger than, inequality. We have further clarified this above and explicitly mentioned that in our proof in the updated manuscript.
> 3. The symbol $\tilde f$ was used in the definition, while $\tilde g$ was used in the statement.
> - For that, we believe it is proper to use different symbols in definitions and theorems, and we have further clarified this above.
>
> Thank you again for your reply. We are always glad to clarify anything you find unclear! You have been very helpful in improving the clarity of our manuscript.
>
> Best,
>
> Submission1650 Authors

---

### Official Review · Reviewer_RnVS · 2023-11-01

**Soundness:** 3 good
**Presentation:** 3 good
**Contribution:** 2 fair
**Rating:** 6
**Confidence:** 3

**Summary:**

The unsupervised probabilistic method for combinatorial optimization is a hot topic in recent machine learning community. The paper proposes UCom2 as a unified framework with principled probabilistic objective construction scheme that provably satisfies the good properties, and a fast and effective de-randomization scheme with a quality guarantee. Under this framework, the paper conduct intensive experiments on combinatorial optimizations with hard constraints and obtains the state-of-the-art performance among unsupervised probabilistic methods.

**Strengths:**

* The paper formally gives principled criteria for objective functions and de-randomization, which completes the framework of unsupervised probabilistic methods for combinatorial optimization. Also, the paper proves the framework is simple but guaranteed to be effective.
* Then paper conduct intensive experiments to empirically demonstrate UCom2 is general and practical. The experiment settings are detailedly provided and the comparison with baselines are properly discussed.

**Weaknesses:**

* Though provides a unified view, the proposed framework is basically doing the same thing as previous methods. Leading the novelty is limited.
* The ad-hoc incremental difference is designed for each problem. Also, only evaluating the difference is a commonly used method to reduce computation. To me, it is more like an engineering effort rather than a machine learning method.
* Minor: the paper looks a bit crowded, making reading a bit tired.

**Questions:**

* I am wondering whether the author tried to compare the performance of UCom2 with supervised learning. I am interested in their gap.
* Since Ucom2 designed a principle de-randomization, I am curious whether it is possible to make the objective function being de-randomization arrogant?

---

> ### Author Response · Authors · 2023-11-14
> **Response to Comment 1: Regarding Novelty**
>
> **Reviewer’s Comment:**
>
> - `Though provides a unified view, the proposed framework is basically doing the same thing as previous methods. Leading the novelty is limited.`
>
> **Response:**
>
> - We believe that there is nothing wrong with “standing on the shoulders of giants”. Although based on existing works, our results are still **theoretically novel** and **empirically useful**.
> - **Our derivations on the complex conditions are nontrivial.** We would like to emphasize our contributions regarding our **derivations** of probabilistic objectives and incremental updates. We first propose two **high-level schemes** (Schemes 1 & 2) in Section 3. Then, in Section 4, following the two high-level schemes, for each considered complex condition, we derive a **probabilistic objective** and **incremental updates** that abide by the proposed schemes. Such derivations (see Lemmas 1-8 and their proofs) are nontrivial, containing solid technical contributions. Specifically, the results by Karalias & Loukas (2020) and Wang et al. (2022) cannot produce such derivations.
> - **Our schemes provide practical general guidelines.** Using the results by Karalias & Loukas (2020) and Wang et al. (2022), when encountering a combinatorial optimization (CO) problem with complex conditions (e.g., any problem considered in this work), one can only know what properties a good probabilistic objective should satisfy, but do not have any practice guideline for how to construct a good one. By contrast, our scheme provides a practical guideline for any problem:
>     - If the encountered problem involves any complex condition considered in this work, our derivations can be immediately used. We have shown the generality of the considered conditions by various problems in Section 5 and Appendix F. That is, we expect that the considered conditions are involved in many problems.
>     - Even if the encountered problem involves some conditions that are not considered in this work, one can still follow our template in Section 4 to construct probabilistic objectives (and incremental updates) for those conditions.
>     - After dealing with each condition involved in the encountered problem, one can always follow our template in Section 5 to obtain a theoretically desirable probabilistic objective (and incremental updates) for the whole problem.
> - **Our derandomization scheme is theoretically and empirically better.** The proposed derandomization scheme achieves a stronger quality guarantee than the existing ones by Karalias & Loukas (2020) and Wang et al. (2022). Specifically, Karalias & Loukas (2020) provided a Markov bound based on random sampling, Wang et al. (2022) provided a deterministic bound based on iterative rounding, and we further improved it with local minimality of the final derandomized decision. See also our ablation study in Appendix G.3.2 comparing our greedy derandomization and iterative rounding by Wang et al. (2022), where we empirically validated the superiority of greedy derandomization over iterative rounding.

---

> ### Author Response · Authors · 2023-11-14
> **Response to Comment 2: Regarding Generality and Significance of Incremental Difference**
>
> **Reviewer’s Comment:**
>
> - `The ad-hoc incremental difference is designed for each problem. Also, only evaluating the difference is a commonly used method to reduce computation. To me, it is more like an engineering effort rather than a machine learning method.`
>
> **Response:**
>
> - We would like to clarify that incremental differences (and probabilistic objectives) are designed for each **complex condition** instead of each problem. For each considered complex condition, we derive a probabilistic objective and incremental updates that abide by the proposed schemes. Such derivations are nontrivial, containing solid **technical contributions**.
>     - See also our response to Comment 1.
> - The complex conditions considered in this work are **commonly involved in many problems** with real-world meanings and applications, as shown in Section 5 and Appendix F.
> - For each problem, we first analyze which complex conditions it involves and then **reuse and combine (instead of constructing from scratch)** the incremental differences (and probabilistic objectives) we have derived for each involved condition.
> - We believe that deriving and using incremental differences are **mathematical and algorithmic** efforts, instead of an “engineering effort”, and deriving the incremental differences per se is a solid **technical contribution**.

---

> ### Author Response · Authors · 2023-11-14
> **Response to Comment 3: Regarding Supervised Learning**
>
> **Reviewer’s Comment:**
>
> - `I am wondering whether the author tried to compare the performance of UCom2 with supervised learning. I am interested in their gap.`
>
> **Response:**
>
> - As discussed in existing works, using supervised learning for combinatorial optimization might encounter the following problems:
>     - it can lead to training instability and/or poor generalization [r1, r2, r3]
>     - it needs labels, i.e., optimal solutions for training instances, which are expensive to obtain, especially when the considered problem is NP-hard and/or the problem instances are large [r1, r3, r4]
> - Even obtaining the “labels” for the training instances would take too long, because all the problems we consider in our experiments are NP-hard, and the scales of our training instances are relatively large. Moreover, even with the labels obtained, the training is expected to be difficult.
> - Indeed, the existing supervised-learning-based methods for combinatorial optimization have been limited to small instances and specific problems (e.g., TSP, routing). Notably, some efforts have been made to improve the generalizability [r5, r6], and we definitely would like to explore supervised learning methods for CO (especially CO with complex conditions) as a future direction.
> - References:
>     - [r1] Karalias and Loukas. "Erdos goes neural: an unsupervised learning framework for combinatorial optimization on graphs." NeurIPS’20
>     - [r2] Joshi et al. "On learning paradigms for the travelling salesman problem." arXiv:1910.07210
>     - [r3] Wang et al. "Unsupervised learning for combinatorial optimization with principled objective relaxation." NeurIPS’22
>     - [r4] Yehuda et al. "It’s not what machines can learn, it’s what we cannot teach." ICML’20
>     - [r5] Fu et al. "Generalize a small pre-trained model to arbitrarily large TSP instances." AAAI’21
>     - [r6] Joshi et al. "Learning the travelling salesperson problem requires rethinking generalization." Constraints 27.1-2 (2022): 70-98.

---

> ### Author Response · Authors · 2023-11-14
> **Response to Comment 4: Regarding Manuscript Density**
>
> **Reviewer’s Comment:**
>
> - `Minor: the paper looks a bit crowded, making reading a bit tiring.`
>
> **Response:**
>
> - Sorry for the inconvenience. We have many contents to be put within the page limit.
> - It might cause confusion if we reorganize the contents during the rebuttal period, but we will try to reorganize the contents for better readability in the camera-ready version.

---

> ### Author Response · Authors · 2023-11-14
> **Response to Comment 5: Regarding Derandomization Arrogant Objectives**
>
> **Reviewer’s Comment:**
>
> - `Since Ucom2 designed a principle de-randomization, I am curious whether it is possible to make the objective function being derandomization arrogant?`
>
> **Response:**
>
> - In our understanding, the reviewer is asking “Can we design objective functions regardless of the derandomization method?”
> - If so, the answer is yes! We are already doing this. Our objective construction scheme does not rely on any specific derandomization method, and the differentiable optimization step is independent of the derandomization step.
> - Further improving the derandomization scheme for UL4CO is actually one of our future directions.
> - Please let us know if we misunderstood the question or if there are still unclear points.

---

> ### Author Response · Authors · 2023-11-20
>
> Dear Reviewer RnVS,
>
> Thank you for your insightful review again!
>
> We are wondering how much our responses have addressed your concerns, and whether you have further questions and comments.
> If you have any specific questions you would like us to elaborate on, please let us know. We value your insights and are open to further discussion to enhance our work.
>
> Thank you once again for your time.
>
> Best,
>
> Submission1650 Authors

---

> ### Comment · Area_Chair_zxxT · 2023-11-22
> **Replies to author comment**
>
> Dear reviewer,
>
> Thank you very much for your work evaluating this review.
>
> It is critical that you urgently address the author's responses, acknowledge their response, and eventually adjust your rating if warranted.
>
> Best,
>
> AC

---

### Official Review · Reviewer_M2Ai · 2023-11-07

**Soundness:** 2 fair
**Presentation:** 3 good
**Contribution:** 2 fair
**Rating:** 3
**Confidence:** 2

**Summary:**

The authors are motivated by the research in unsupervised combinatorial optimization and propose to extend prior work in this topic to more complex optimization problems. In particular, they seek to develop a principled approach to constructing probabilistic objectives and effective derandomization scheme with guarantee on solution quality.

**Strengths:**

The ideas around unsupervised combinatorial optimization are interesting and authors place their contribution well by discussing prior results and adequately motivating their work. The theoretical quality guarantee from derandomization scheme seems to be a reasonable contribution.

**Weaknesses:**

Since the paper builds on the specific work Karalias & Loukas (2020) and Wang et al. (2022), I was not able to get a solid understanding of conceptual contributions of the paper. Theoretical results - as claimed by the authors - are follow fairly standard arguments. They are based on standard (basic) optimization analysis. While the tightness of these results to the original approach proposed by  Karalias & Loukas might be worthwhile, we do not get a sufficient understanding of the generality of these results. I would have appreciated seeing a more pointed discussion on why rounding/derandomization schemes in classical combinatorial optimization are not helpful here? Without such a consideration, the contribution might myopically advance the idea of pushing differentiable optimization into combinatorial optimization, but might miss on building on rich set of existing results in combinatorial optimization.

**Questions:**

- how does the result on goodness of greedy derandomization compare with similar ideas in combinatorial optimization?
- what do we mean by problems with "complex conditions"?
- what are the features of the problems studied in the experiments section that enable a good solution guarantee after derandomization (vs not)?

---

> ### Author Response · Authors · 2023-11-14
> **Response to Comment 1: Regarding “Complex Conditions”**
>
> **Reviewer’s Comment:**
>
> - `What do we mean by problems with "complex conditions"?`
>
> **Response:**
>
> - As mentioned in the introduction, in this work, “complex conditions” include **complex optimization objectives and/or constraints that are mathematically hard to handle**.
> - More specifically, for such conditions, **deriving probabilistic objectives and incremental updates is nontrivial**, which is one of our main conceptual and technical contributions.
> - It is **non-trivial to formally define** “complex conditions”, but we can summarize some **common characteristics** of the complex conditions considered in this work, i.e., the considered complex conditions usually involve a combination of **multiple dependent** sub-events:
>     - **Cardinality constraints:** the selection of multiple elements, where the dependency lies in that if one element is not selected, another element can be selected without the cardinality unchanged, i.e., whether an element can be selected without violating the constraints depends on how many other elements are selected.
>     - **Optimum w.r.t. a subset:** the optimum of multiple choices in the subset, where the dependency lies in that an element in the subset is optimal (and thus used to compute the objective) only if all the better elements are not chosen.
>     - **Covered:** the coveredness of multiple elements in the considered set.
>     - **Cliques:** the adjacency between each chosen node pair, where the dependency lies in that, whether a node violates the constraints depends on whether other nodes that are not adjacent to it are chosen.
>     - **Non-binary decisions:** in general, the number of sub-events increases significantly with non-binary decisions, e.g., when the number of decisions increases from 2 to 4, the number of possible sub-events gets squared ($2^n \to 4^n$).
>     - **Uncertainty:** in general, with intrinsic uncertainty in the problem, we need to consider multiple possible scenarios even when we fix decisions, which increases the total number of possible sub-events we need to consider.
>     - One direct consequence is that it is **non-trivial** to construct the corresponding probabilistic objectives by **enumerating** all possible situations, which often requires $\Omega(2^n)$ evaluations of the objective function, and the number of required evaluations is even higher with non-binary decisions and/or uncertainty.
> - We show **various examples** of complex conditions (see Sec. 4) and problems involving such conditions (see Sec. 5 and Appendix F).

---

> ### Author Response · Authors · 2023-11-14
> **Response to Comment 2: Regarding Existing Techniques in Classical Combinatorial Optimization**
>
> **Reviewer’s Comment:**
>
> - `I would have appreciated seeing a more pointed discussion on why rounding/derandomization schemes in classical combinatorial optimization are not helpful here. Without such a consideration, the contribution might myopically advance the idea of pushing differentiable optimization into combinatorial optimization, but might miss on building on the rich set of existing results in combinatorial optimization.`
> - `How does the result on the goodness of greedy derandomization compare with similar ideas in combinatorial optimization?`
>
> **Response:**
>
> - Thanks for the advice on enriching the discussions on the connections to the existing rounding/derandomization techniques in classical combinatorial optimization (CO). Such discussions have been indeed missing in this line of research on probabilistic-method-based UL4CO.
> - We found that the existing rounding/derandomization techniques in classical CO, especially randomized rounding, are **actually helpful** and have been **implicitly generalized by existing works**.
>     - The **sampling** in the result by Karalias & Loukas (2020) (see Theorem 5 in Appendix A.3) can be seen as a generalization of **independent randomized rounding**, where each entry is derandomized by an independent Bernoulli trial.
>     - The **iterative rounding** by Wang et al. (2022) (see Theorem 6 in Appendix A.3) can be seen as a generalization of randomized rounding using **the method of conditional probabilities**.
> - We **could not find** existing randomized rounding techniques similar to our derandomization scheme, which implies that the idea in our derandomization scheme might be even useful in randomized rounding for classical CO.
> - Regarding the question `How does the result on the goodness of greedy derandomization compare with similar ideas in combinatorial optimization?`, as mentioned above, we believe that some ideas of randomized rounding **have been implicitly generalized** by Karalias & Loukas (2020) and Wang et al. (2022), while as shown in this work, the proposed greedy derandomization scheme achieves a stronger theoretical guarantee and better empirical performance.
> - We **could not find** other advanced rounding/derandomization schemes that are directly applicable to our case. It would be highly appreciated if the reviewer could suggest some specific rounding/derandomization schemes in classical combinatorial optimization we can consider and apply.
> - Again, we would like to emphasize our contributions regarding our nontrivial **derivations** of probabilistic objectives and incremental updates. For classical CO methods using integer/linear programming, the objective construction/relaxation is usually done by simply relaxing the range of variables without changing the formula. Such a naive way might violate the good properties (see Target 1 in Section 3.1) in the UL4CO setting, and can be improper.
> - We believe that the solutions obtained by our method (or machine learning for CO in general) **can be immediately helpful for classical CO**, e.g., providing initial solutions, and fast evaluation. Also, we definitely would like to explore the possibilities of incorporating existing classical CO techniques into differentiable combinatorial optimization.

---

> > ### Comment · Reviewer_M2Ai · 2023-11-14
> >
> > Thanks for the clarification.

---

> > > ### Author Response · Authors · 2023-11-17
> > > **RE: Official Comment by Reviewer M2Ai**
> > >
> > > Dear Reviewer M2Ai,
> > >
> > > Thank you for your kind reply!
> > >
> > > We are wondering how much our responses have addressed your concerns, and whether you have further questions and comments.
> > > If you have any specific questions you would like us to elaborate on, please let us know. We value your insights and are open to further discussion to enhance our work.
> > >
> > > Thank you once again for your time.
> > >
> > > Best,
> > >
> > > Submission1650 Authors

---

> ### Author Response · Authors · 2023-11-14
> **Response to Comment 3: Regarding Conceptual Contributions**
>
> **Reviewer’s Comment:**
>
> - `Since the paper builds on the specific work by Karalias & Loukas (2020) and Wang et al. (2022), I was not able to get a solid understanding of the conceptual contributions of the paper. Theoretical results - as claimed by the authors - follow fairly standard arguments. They are based on standard (basic) optimization analysis. While the tightness of these results to the original approach proposed by Karalias & Loukas might be worthwhile, we do not get a sufficient understanding of the generality of these results.`
>
> **Response:**
>
> - The reviewer seemingly focused on greedy derandomization *per se* as an isolated component. We would like to clarify that
>   - The goodness of greedy derandomization relies on the goodness of probabilistic objectives (specifically, the entry-wise concavity of the probabilistic objective, which is ensured by our principled construction), i.e., "how to construct good probabilistic objectives such that greedy derandomization works well on them" is nontrivial. Overall, **Schemes 1 and 2 synergize as a whole** and we should see the values of each scheme considering that they complement each other.
>   - Our derivations on each complex condition are nontrivial, which we will elaborate on below.
> - **Our derivations on the complex conditions are nontrivial.** We would like to emphasize our contributions regarding our **derivations** of probabilistic objectives and incremental updates. We first propose two **high-level schemes** (Schemes 1 & 2) in Section 3. Then, in Section 4, following the two high-level schemes, for each considered complex condition, we derive a **probabilistic objective** and **incremental updates** that abide by the proposed schemes. Such derivations (see Lemmas 1-8 and their proofs) are nontrivial, containing solid technical contributions. Specifically, the results by Karalias & Loukas (2020) and Wang et al. (2022) cannot produce such derivations.
> - **Our schemes provide practical general guidelines.** Using the results by Karalias & Loukas (2020) and Wang et al. (2022), when encountering a combinatorial optimization (CO) problem with complex conditions (e.g., any problem considered in this work), one can only know what properties a good probabilistic objective should satisfy, but do not have any practice guideline for how to construct a good one. By contrast, our scheme provides a practical guideline for any problem:
>     - If the encountered problem involves any complex condition considered in this work, our derivations can be immediately used. We have shown the generality of the considered conditions by various problems in Section 5 and Appendix F. That is, we expect that the considered conditions are involved in many problems.
>     - Even if the encountered problem involves some conditions that are not considered in this work, one can still follow our template in Section 4 to construct probabilistic objectives (and incremental updates) for those conditions.
>     - After dealing with each condition involved in the encountered problem, one can always follow our template in Section 5 to obtain a theoretically desirable probabilistic objective (and incremental updates) for the whole problem.
> - **Our derandomization scheme is theoretically and empirically better.** As the reviewer pointed out, the proposed derandomization scheme achieves a stronger quality guarantee than the existing ones by Karalias & Loukas (2020) and Wang et al. (2022). Specifically, Karalias & Loukas (2020) provided a Markov bound based on random sampling, Wang et al. (2022) provided a deterministic bound based on iterative rounding, and we further improved it with local minimality of the final derandomized decision. See also our ablation study in Appendix G.3.2 comparing our greedy derandomization and iterative rounding by Wang et al. (2022), where we empirically validated the superiority of greedy derandomization over iterative rounding.
> - **Our theoretical results are nontrivial and general.**
>     - Theorem 1 (principled objective construction) is nontrivial, where we show that the expectation of **any discrete function** is differentiable and entry-wise concave, which enables us to principally construct good probabilistic objectives (see Scheme 1).
>     - Theorem 2 (goodness of greedy derandomization) is nontrivial. The statement depends on a desirable probabilistic objective, which is guaranteed by our objective construction scheme. This shows the **synergy** between our two proposed schemes.
>     - Theorems 3 and 4 regarding problems with non-binary decisions are novel, since **non-binary decisions were not discussed** in the existing works by Karalias & Loukas (2020) and Wang et al. (2022).
>     - Our main theoretical results (Theorems 1-4) are **general**, holding true **for any complex condition** and thus for any problem involving such conditions.

---

> ### Author Response · Authors · 2023-11-14
> **Response to Comment 4: Regarding the Features of the Problems Enabling a Good Solution Guarantee**
>
> **Reviewer’s Comment:**
>
> - `What are the features of the problems studied in the experiments section that enable a good solution guarantee after derandomization (vs not)?`
>
> **Response:**
>
> - The **main theorems** on the goodness of greedy derandomization (Theorems 2 and 4) hold true **for any problem**.
>     - **For any problem**, the derandomized decision is guaranteed to be no worse than the initial probabilistic decision obtained by differentiable optimization. See (G2) in Theorem 2.
>     - **For any problem**, the derandomized decision is guaranteed to be a local minimum. See (G3) in Theorem 2.
> - We chose the problems because they **involve the complex conditions** we considered, and have **real-world meanings and applications**.
> - We **did not choose problems in favor of our method**. Notably, discrete greedy algorithms are used as a baseline method in each problem in our experiments and the proposed method often outperforms greedy algorithms.

---

> ### Author Response · Authors · 2023-11-23
>
> Dear Reviewer M2Ai,
>
> Thank you for your thorough review of our paper again.
>
> Since the discussion phase is close to the end, we would like to inquire if our responses have addressed your concerns, and if so, whether you could consider changing your rating.
> - The reviewer seemingly focused on greedy derandomization *per se* as an isolated component. We would like to clarify that (1)  "how to construct good probabilistic objectives such that greedy derandomization works well on them" is nontrivial (Theorem 2 requires the entry-wise concavity of the objective ensured by Theorem 1; Schemes 1 and 2 synergize as a whole), and (2) our derivations for each condition are nontrivial (Lemmas 1-8).
> - Please check our responses above for more details.
>
> We remain fully committed to addressing any questions you may have by the end of the discussion phase.
>
> We sincerely appreciate your time and effort in reviewing our paper. We eagerly await your response!
>
> Best,
>
> Submission1650 Authors